# Solvent vapor diffusion–driven multiscale pre-aggregation of non-fullerene acceptors enables high-performance organic solar cells

Weilin Zhou[1], Xingjian Dai[1], Ben Fan[1], Hongxiang Li[2], Xiaopeng Xu[1] ✉, Yihui Wu[1] & Qiang Peng[1] ✉

Precise control of active layer morphology is crucial for improving organic solar cell efficiency but remains challenge. Here, we report a solvent vapor diffusion (SVD) method that creates a vertical solvent gradient by diffusing benzene vapor into a toluene-based acceptor solution before layer-by-layer fabrication. This process tunes multiscale pre-aggregation of non-fullerene acceptors, forming hierarchical domains that facilitate efficient exciton dissociation and charge transport. This strategy also optimizes the vertical donor/acceptor composition profile, aligning exciton generation to minimize recombination and enhance charge collection. Morphological, spectroscopic, and device analyses show enhanced molecular ordering, improved phase separation, and superior carrier dynamics compared with conventional processing. Solvent vapor diffusion strategy demonstrates universality across diverse systems, yielding consistent performance gains. Devices based on D18/L8-BO achieved an efficiency of 20.18%, while D18 (1% PYIT)/L8-BO-C4 reached 20.71%, establishing the strategy as a powerful approach for structural engineering in high-performance OSCs.

Organic solar cells (OSCs) have recently attracted substantial attention owing to their lightweight nature, mechanical flexibility, semitransparency, and compatibility with cost-effective, scalable manufacturing[1–5]. The emergence of non-fullerene acceptors (NFAs) has propelled power conversion efficiencies (PCEs) of state-of-the-art OSCs beyond 20%[6–13], benefiting from their chemical tunability and strong light-harvesting[14,15]. However, robust control of photoactive layer morphology remains a major challenge, as it critically impacts exciton dissociation, charge transport, and recombination[16–19]. Crucially, solution-to-film assembly pathways shape nanostructure and optoelectronic properties[20,21], where π–π stacking, dispersion, and van der Waals interactions drive aggregation and charge-transport outcomes[19]. In bulk heterojunction (BHJ) systems, morphology is determined by the coupled self-assembly of donors and acceptors, their miscibility and mutual aggregation, and distinct crystallization kinetics[22], with liquid–liquid and solid–liquid phase separation co-

dictating nanoscale structure[23]. As a result, multicomponent solutions exhibit complex interaction and divergent crystallization behaviors that are highly sensitive to processing conditions, hindering reproducible, optimal BHJ morphologies. Layer-by-layer (LbL) processing addresses these limitations by sequentially depositing donor and acceptor layers, decoupling their aggregation and phase behaviors and enabling precise control of microstructure and interfaces[24–26]. This strategy allows fine-tuning of vertical composition and domain architecture—features that are difficult to regulate in one-step BHJ processing[27–31]. It also facilitates systematic probing of how solution pre-aggregation and interfacial assembly impact morphology and charge transfer. Nevertheless, reliably directing the pre-aggregation of both components to achieve targeted nanoscale structures remains challenging.

Solvent vapor diffusion (SVD), a classic technique in materials science and organic electronics, has been widely used to grow high-

[1]School of Chemical Engineering and State Key Laboratory of Advanced Polymer Materials, Sichuan University, Chengdu 610065, P. R. China. [2]College of Polymer Science and Engineering, Sichuan University, Chengdu 610065, P. R. China. ✉e-mail: xpxu@scu.edu.cn; qiangpeng@scu.edu.cn

quality single crystals of organic semiconductors[32–35]. SVD leverages controlled antisolvent exposure to induce supersaturation, balancing nucleation and crystal growth to facilitate thermodynamically favored assembly. This process not only enables the formation of optimal aggregate structures but can also circumvent kinetic trapping, which often leads to undesirable morphologies. While SVD has predominantly been applied to crystal growth, its utility in regulating the pre-aggregation of NFAs for OSC applications remains largely unexplored. Harnessing SVD to pre-assemble acceptor molecules may offer opportunities to tune phase separation and optimize charge generation and transport in OSC active layers.

In this work, we present an SVD-guided strategy to precisely regulate the multiscale pre-aggregation of NFAs for efficient LbL fabrication of OSCs. Using D18 as the donor polymer and L8-BO as the model acceptor, we first spin-coat the D18 layer from chloroform, then expose a toluene solution of L8-BO to benzene vapor for controlled durations before depositing it onto the D18 layer. The SVD process establishes a vertical supersaturation gradient within the L8-BO solution: the benzene-rich upper region promotes strong pre-aggregation of L8-BO molecules, whereas the toluene-rich lower region preserves weaker pre-aggregation. This produces a well-defined top-down gradient that is transferred to the active layer during film formation. The resulting hierarchical assembly enables favorable phase separation across multiple length scales and enhances crystallinity in the final films. Compared with established morphology-control strategies such as ternary blends, additives, and solution aging[36–39], SVD offers distinctive advantages: (1) it creates vertically graded, multiscale aggregates that co-optimize charge generation and transport; (2) its simple, self-limiting process eliminates additive-specific screening while delivering reproducibility; and (3) it integrates readily with complementary treatments to achieve finer morphological control. Leveraging these multiscale optimizations, the PCE increases from 18.99% (without SVD) to 20.18% for D18/L8-BO, and further to 20.71% for D18 (1% PYIT)/L8-BO-C4. The universality of the approach is further validated across additional acceptor systems (BTP-C6, m-THE, eC9-2F2Cl) and the donor PM6, confirming its broad applicability.

Collectively, these results establish SVD as a robust, versatile method for controlling vertical aggregate distributions in LbL-OSCs and as a powerful platform for advancing fundamental understanding of solution-to-film assembly, opening new design space and efficiency ceilings for next-generation organic photovoltaics.

## Results

### Solvent vapor diffusion processing details

Figure 1a presents the chemical structure of the L8-BO acceptor, which exhibits significantly higher solubility in toluene than in benzene (17.3 vs. 9.1 mg/mL; Supplementary Fig. 1). Figure 1b schematically illustrates the SVD setup, where a small, lidless bottle containing the NFA solution (L8-BO in toluene) is placed inside a larger sealed bottle containing benzene, a more volatile solvent than toluene (saturated vapor pressure at 20 °C: 12.8 vs. 2.4 kPa). As a result, the vapor phase inside the large bottle is dominated by benzene. During the SVD process, benzene vapor diffuses into the small bottle and condenses as a liquid at the surface of the NFA solution, then gradually diffuses from top to bottom, establishing a vertical concentration gradient of benzene and toluene as simulated by COMSOL (Fig. 1c; Supplementary Fig. 2). Here, SVD-x denotes the L8-BO solution treated with SVD for x minutes, where x = 0 represents the untreated control. The benzene fraction within the L8-BO toluene solution is highly dependent on both solution depth and SVD duration (Fig. 1d), giving rise to three main regions: region A—the benzene-rich upper layer; region B—the intermediate region with nearly equal proportions of benzene and toluene; and region C—the toluene-rich bottom layer. This clearly demonstrates a hierarchical distribution of the two solvents. As benzene is a poorer solvent for L8-BO compared to toluene, this solvent gradient promotes graded pre-aggregation of L8-BO from top to bottom. With increasing SVD duration (0 to 120 min), regions A and B expand while region C shrinks, indicating that the vertical solvent environment can be effectively tuned by adjusting the SVD time. Meanwhile, the overall benzene fraction in the solution gradually increases from 0% to 55% as SVD time extends to 120 min (Fig. 1e), with simulated benzene fractions closely matching experimental results (Supplementary Fig. 3;

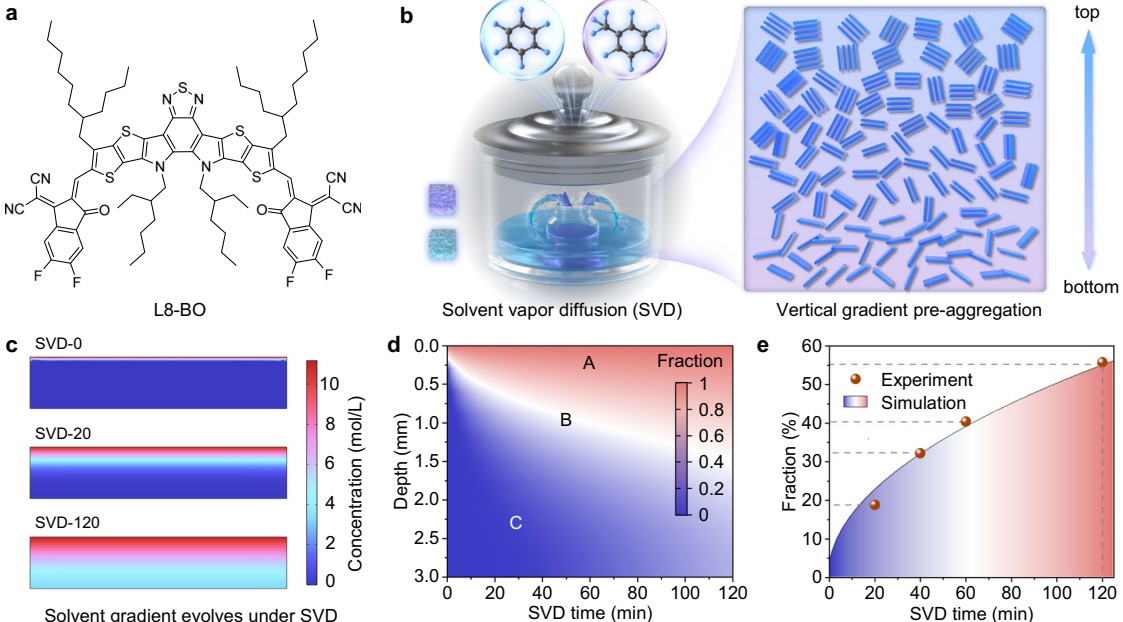

**Fig. 1 | Molecular structure and solvent vapor diffusion method. a** Chemical structure of L8-BO. **b** Schematic illustration of the SVD process enabling vertical gradient control of acceptor pre-aggregation in solution. **c** COMSOL simulation showing benzene concentration gradient evolution during SVD. **d** Simulated spatial distribution of benzene/toluene fraction in solution as a function of SVD time. **e** Simulated and experimental evolution of benzene fraction in solution as a function of SVD time.

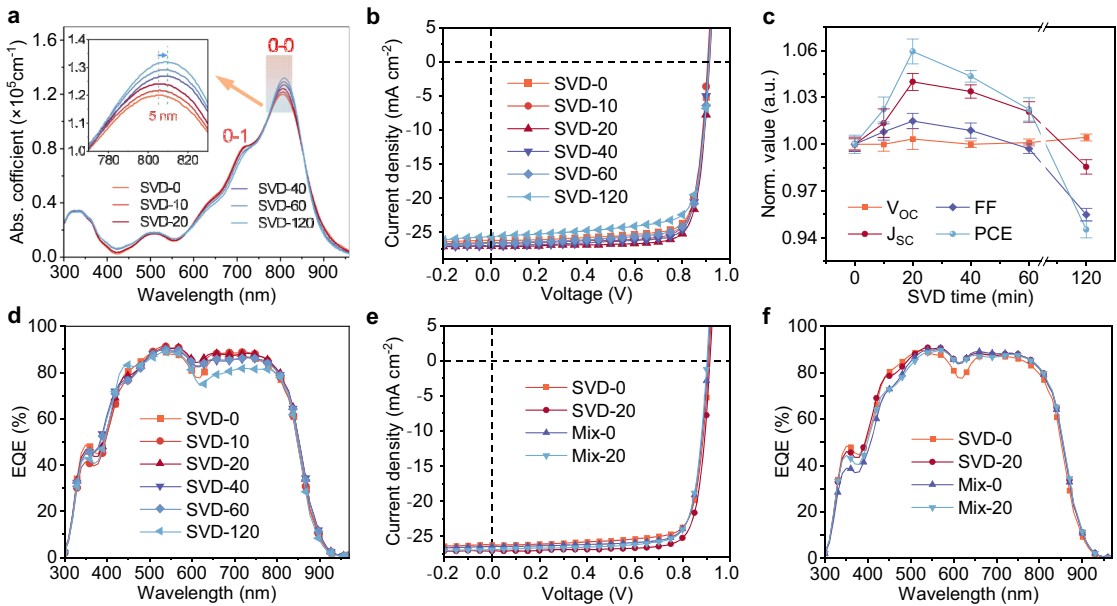

**Fig. 2 | Device performance of D18/L8-BO based OSCs. a** Absorption coefficient spectra of L8-BO films prepared from solutions with different SVD durations. **b** *J-V* curves of OSCs fabricated with L8-BO subjected to various SVD durations. **c** Normalized $V_{OC}$, $J_{SC}$, FF, and PCE as a function of SVD duration. Each parameter was calculated from 10 individual devices. **d** EQE spectra of OSCs fabricated with L8-BO subjected to different SVD durations. **e** *J-V* curves of OSCs fabricated with L8-BO treated via Mix-0, Mix-20, SVD-0, and SVD-20 processes. **f** EQE spectra of OSCs fabricated with L8-BO treated via Mix-0, Mix-20, SVD-0, and SVD-20 processes.

**Table 1 | Photovoltaic parameters of D18/L8-BO based OSCs fabricated using L8-BO subjected to SVD treatment with different durations before deposition on the D18 donor layer**

| SVD condition | $V_{OC}$[a] [V] | $J_{SC}$[a] [mA cm$^{-2}$] | $J_{EQE}$[b] [mA cm$^{-2}$] | FF[a] [%] | PCE[a] [%] |
|---|---|---|---|---|---|
| SVD-0 | 0.910 | 26.21 | 25.54 | 79.63 | 18.99 |
| | 0.908 ± 0.003 | 26.06 ± 0.12 | | 79.74 ± 0.31 | 18.86 ± 0.11 |
| SVD-10 | 0.906 | 26.54 | 25.91 | 80.77 | 19.43 |
| | 0.908 ± 0.004 | 26.41 ± 0.24 | | 80.37 ± 0.42 | 19.28 ± 0.15 |
| SVD-20 | 0.915 | 27.08 | 26.20 | 81.47 | 20.18 |
| | 0.911 ± 0.006 | 27.10 ± 0.14 | | 80.93 ± 0.39 | 19.98 ± 0.15 |
| SVD-40 | 0.909 | 26.99 | 25.89 | 80.48 | 19.75 |
| | 0.908 ± 0.002 | 26.94 ± 0.11 | | 80.44 ± 0.39 | 19.68 ± 0.07 |
| SVD-60 | 0.911 | 26.63 | 25.84 | 79.70 | 19.34 |
| | 0.909 ± 0.002 | 26.65 ± 0.17 | | 79.52 ± 0.24 | 19.28 ± 0.14 |
| SVD-120 | 0.913 | 25.66 | 24.85 | 76.69 | 17.96 |
| | 0.912 ± 0.002 | 25.68 ± 0.12 | | 76.15 ± 0.32 | 17.83 ± 0.10 |

[a]The averaged values with standard deviations are calculated from ten individual devices. [b]Short-circuit current density values are calculated from EQE integration.

Supplementary Table 1). Because benzene is more volatile than toluene, a higher benzene fraction in the solution accelerates film drying during deposition. Under these conditions, not only can the graded pre-aggregation of L8-BO in solution be precisely tuned, but the subsequent film formation dynamics can also be optimized via SVD, potentially achieving well-controlled, multiscale domain structures of L8-BO within the photoactive layer.

UV-vis-NIR absorption measurements confirm the enhanced molecular packing order in L8-BO films prepared from solutions with different SVD durations (Fig. 2a; Supplementary Table 2). As the SVD time increases to 120 min, the maximum absorption peak exhibits a gradual redshift of 5 nm (from 805 to 810 nm), accompanied by increased absorption coefficient (from $1.20 \times 10^5$ to $1.32 \times 10^5$ cm$^{-1}$). Moreover, the intensity ratio of $I_{0-0}$ (0-0 electronic-vibration transition) to $I_{0-1}$ (0-1 electronic-vibration transition) for L8-BO increases progressively (from 1.48 to 1.68) with SVD duration, indicating a steady

improvement in molecular order and stronger *J*-aggregate characteristics in the solid state.

**Photovoltaic performance**

To investigate the effect of SVD duration on photovoltaic performance, LbL-OSCs were fabricated with the structure of ITO/Ph-4PACz/D18/L8-BO/PNDIT-F3N/Ag. L8-BO solutions were subjected to SVD treatment for varying durations prior to sequential deposition onto the D18 (chemical structure in Supplementary Fig. 4) layer, forming the photoactive layer. Representative current density–voltage (*J–V*) characteristics are presented in Fig. 2b, and the statistical photovoltaic parameters for devices treated with different SVD durations are summarized in Table 1.

As a solvent control, devices processed from benzene (without SVD) yielded a PCE of 18.44% with open-circuit voltage ($V_{OC}$) of 0.924 V, short-circuit current density ($J_{SC}$) of 26.48 mA cm$^{-2}$, and fill

factor (FF) of 75.37% (Supplementary Fig. 5). In contrast, the toluene-processed control (SVD-0, toluene) achieved a higher PCE of 18.99% with $V_{OC}$ of 0.910 V, $J_{SC}$ of 26.21 mA cm$^{-2}$, and FF of 79.63%, indicating that toluene provided a more favorable morphology baseline despite its lower $V_{OC}$. While the $V_{OC}$ remained largely insensitive to SVD, the $J_{SC}$ and FF depended strongly on duration, markedly influencing PCE (Fig. 2c). SVD-10 yielded a modest gain (PCE = 19.43%, $J_{SC}$ = 26.54 mA cm$^{-2}$, FF = 80.77%), and SVD-20 gave the best performance (PCE = 20.18%, $J_{SC}$ = 27.08 mA cm$^{-2}$, FF = 81.47%). SVD-40 maintained a similar high $J_{SC}$ (26.99 mA cm$^{-2}$) with slightly reduced FF (80.48%), leading to a PCE of 19.75%. SVD-60 showed modest declines in both $J_{SC}$ (26.63 mA cm$^{-2}$) and FF (79.70%), with PCE dropping to 19.34%. Importantly, SVD for 10–60 min consistently outperformed SVD-0, indicating a broad processing window. Excessive SVD proved detrimental. SVD-120 reduced $J_{SC}$ (25.66 mA cm$^{-2}$) and FF (76.69%), lowering PCE to 17.96%, likely due to over-aggregation or unfavorable phase separation from an overly steep solvent gradient and prolonged reorganization.

To isolate SVD effects from additive influences, 1,4-diiodobenzene (DIB) additive-free devices were fabricated while keeping all other conditions unchanged. Although absolute efficiencies were slightly lower, the performance dependence on SVD duration was preserved, confirming the intrinsic effectiveness of SVD (Supplementary Fig. 6; Supplementary Table 3). DIB remains fully dissolved in both solvents at concentration of 50 mg/mL (Supplementary Fig. 7), well above the 15 mg/mL used in our SVD experiments. Consequently, unlike L8-BO, DIB does not undergo solubility-driven pre-aggregation in either solvent during SVD and thus has minimal influence on the SVD process.

To further elucidate the impact of SVD duration on device performance, external quantum efficiency (EQE) spectra were measured (Fig. 2d; Table 1). Devices with intermediate SVD durations showed enhanced EQE responses in both donor and acceptor absorption regions, with especially notable improvement in the acceptor region (600–800 nm), indicating more efficient charge generation and transport. However, prolonged SVD reduced the L8-BO concentration in the solution—and thus its proportion in the final photoactive layer film (Supplementary Fig. 8)—thereby diminishing the EQE response in the acceptor region.

To rigorously assess the advantage of the SVD-induced vertical solvent gradient, we conducted two mixing controls: Mix-0, in which the same amount of benzene as in SVD-20 was added to the L8-BO toluene solution, thoroughly mixed, and immediately cast, and Mix-20, in which the same benzene amount was thoroughly mixed and allowed to stand for 20 minutes before casting. The resulting PCEs were similar for Mix-0 (19.23%) and Mix-20 (19.33%), higher than SVD-0 (pure toluene) but still below SVD-20, despite matched benzene content (Fig. 2e, f; Supplementary Table 4). These results indicate that the vertical solvent gradient established by SVD affords additional morphological and performance advantages beyond those attainable by homogeneous solvent mixing.

**Thin film forming dynamics and morphology**

The effect of SVD on film formation dynamics was probed by in-situ absorption, monitoring L8-BO during spin coating onto the D18 layer (Supplementary Fig. 9). The absorption spectra delineate three stages: Stage I, solvent evaporation, where the absorption peak position remains nearly unchanged; Stage II, nucleation and crystal growth, where supersaturation triggers rapid self-assembly and crystallization; Stage III, film drying, where the peak stabilizes with no further significant changes. In Stage II, a key kinetic descriptor is the crystal-growth duration $\Delta t$, which shortens with increasing SVD time (from 1.07 s for SVD-0 to 0.56 s for SVD-120) (Fig. 3a; Supplementary Fig. 10). This acceleration correlates with the higher benzene fraction at longer SVD durations, given benzene's lower solubility and higher volatility, which promote faster supersaturation and growth. Notably, while SVD-

20, Mix-0, and Mix-20 have the same overall benzene fraction, their $\Delta t$ values differ: 0.92 s (SVD-20), 1.04 s (Mix-0), and 1.02 s (Mix-20) (Fig. 3b). This divergence indicates that SVD, by establishing a vertical concentration gradient between two solvents with disparate solubilities, induces stronger and spatially biased pre-aggregation of L8-BO than achievable by homogeneous mixing, thereby enabling faster and more directed crystallization during Stage II.

The impact of SVD on film packing order was investigated using grazing-incidence wide-angle X-ray scattering (GIWAXS) (Supplementary Fig. 11; Supplementary Table 5). Neat D18 exhibits both lamellar stacking and π-π stacking in the in-plane (IP) and out-of-plane (OOP) directions, suggesting coexistence of face-on and edge-on orientations. In comparison, D18/L8-BO blend films show a pronounced π-π stacking signal primarily along OOP, indicative of a predominantly face-on packing. For SVD-0, a sharp lamellar peak appears at $q_{xy}$ = 0.313 Å$^{-1}$ (lamellar spacing $d_l$ = 20.1 Å, crystal coherence length CCL = 104 Å) in IP, while the π-π peak appears at $q_z$ = 1.67 Å$^{-1}$ (π-π spacing $d_{\pi-\pi}$ = 3.77 Å, CCL = 35.8 Å) in OOP. Increasing SVD time leads to a progressive increase in CCL for both lamellar stacking (from 104 to 156 Å) and π-π stacking (from 35.8 to 42.6 Å), demonstrating enhanced molecular order.

A comparative analysis of SVD-20, Mix-0, and Mix-20 films (Supplementary Fig. 12; Supplementary Table 6) reveals similar peak positions and stacking distances, yet SVD-20 achieves larger CCLs (IP: 138 Å, OOP: 44.5 Å) than Mix-0 (IP: 134 Å, OOP: 40.7 Å) and Mix-20 (IP: 135 Å, OOP: 42.2 Å), and all exceed SVD-0 (IP: 108 Å, OOP: 35.7 Å) (Fig. 3c). These results suggest SVD promotes L8-BO pre-aggregation and the formation of a more ordered crystalline phase compared to that of homogeneous solvent mixing.

Atomic force microscopy (AFM) was used to probe the photoactive layer morphology (Supplementary Fig. 13). All films display distinct fibrous network structures. With increasing SVD time, the root-mean-square (RMS) roughness increases steadily (from 1.09 to 1.99 nm), indicating strengthened phase separation. Notably, the RMS roughness of SVD-20 (1.52 nm) surpasses that of Mix-0 (1.22 nm) and Mix-20 (1.33 nm), aligning with the stronger enhancement of L8-BO crystallinity observed by GIWAXS.

To further quantify phase separation, grazing-incidence small-angle X-ray scattering (GISAXS) was performed (Supplementary Fig. 14). Line cuts along the $q_r$ (IP direction) were fitted using the Debye-Anderson-Brumberger (DAB) and fractal models with SasView. Key parameters (Supplementary Table 7; Fig. 3c) include donor-phase average correlation length ($\xi$), acceptor-phase correlation length ($\eta$) and fractal dimension ($D$). The acceptor domain size was estimated as $2R_g$ calculated from $\eta$ and $D$. With longer SVD time, $\xi$ remains nearly constant, but $2R_g$ increases substantially—from 22.9 nm (SVD-0) to 43.1 nm (SVD-20) and 64.9 nm (SVD-120)—indicating significant domain coarsening. Notably, SVD-20 also yields a larger $2R_g$ than Mix-0 (32.86 nm) and Mix-20 (37.86 nm), consistent with the formation of multiscale L8-BO domains under the vertical solvent-gradient environment of SVD. Such multiscale domains can, in principle, provide both abundant donor/acceptor (D/A) interfaces for efficient charge separation (smaller domains) and continuous pathways that favor charge transport (larger domains).

Grazing-incidence transmission small-angle X-ray scattering (GTSAXS) was employed to resolve the nanoscale morphology at different depths of the active layer. Incidence angles of 0.08°, 0.12°, 0.16°, and 0.20° were used to probe the surface, upper-middle, middle, and lower-middle regions[40,41], respectively (Supplementary Figs. 15–18). To reconstruct the three-dimensional (3D) nanoscale morphology, we quantified the $2R_g$ and the extent of intermixed phases ($\varphi$) by fitting the IP line cuts with the DAB and fractal models. Both $2R_g$ and $\varphi$ exhibit negligible variation with angle for SVD-0 and Mix-20, although their absolute values in Mix-20 are slightly higher than in SVD-0 (Fig. 3d; Supplementary Table 8). This increase is attributed to benzene-

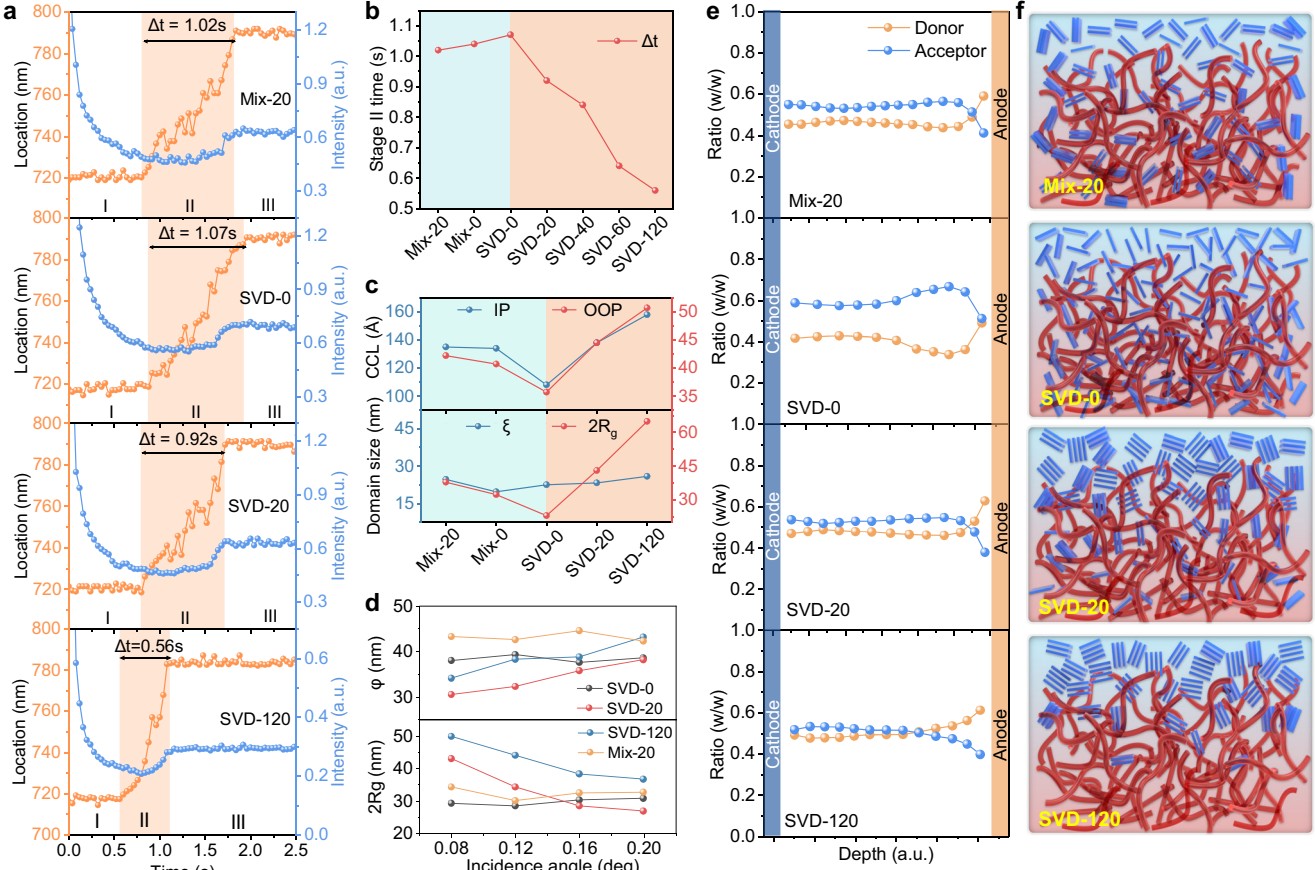

**Fig. 3 | Molecular crystallinity and film morphology. a** Evolution of the absorption peak position and intensity of L8-BO extracted from 2D time-resolved in-situ absorption spectra under various processing conditions. **b** Stage II times extracted from the time-resolved in-situ absorption spectra. **c** Coherence crystal length (CCL) calculated from GIWAXS and domain sizes of donor and acceptor phases obtained from GISAXS. **d** Domain sizes of acceptor phases ($2R_g$) and intermixed phases ($\varphi$) determined from GTSAXS. **e** Component distribution profiles at different depths within the photoactive layers. **f** Schematic illustration of the vertical phase separation in various photoactive layers. (Red fibers represent polymer donor D18, and blue fibers indicate the acceptor L8-BO).

induced aggregation of the acceptor. In contrast, SVD-20 and SVD-120 show a depth dependence: the $2R_g$ decreases progressively with increasing angle, with SVD-120 presenting a steeper decline. Both SVD-20 and SVD-120 display larger $\varphi$ at 0.12° and 0.16°, indicating a higher density of acceptor-related mixed interfaces in the central region of the active layer. At 0.20°, $\varphi$ for SVD-20 exceeds that of SVD-120. Below the middle region, the fraction of donor-rich interfaces in SVD-120 diminishes more rapidly, which is consistent with reduced exciton dissociation relative to SVD-20. Taken together, the GTSAXS results reveal a vertically graded distribution of the donor and acceptor phases in SVD-treated films. The SVD-induced gradient pre-aggregation of the acceptor modulates the vertical morphology of the active layer and establishes depth-dependent mixed interfaces.

Film-depth-dependent absorption spectroscopy (FLAS) was employed to investigate the vertical compositional distribution of D18/L8-BO blends. While the characteristic absorption peaks of D18 (500–600 nm) and L8-BO (750–850 nm) remain consistent across depth (Supplementary Fig. 19), the vertical D/A compositional profiles vary with SVD duration (Fig. 3e; Supplementary Fig. 20). Notably, the overall fraction of L8-BO decreases with longer SVD durations, attributable to the progressive diffusion of volatile benzene into the L8-BO toluene solution during SVD, which dilutes the acceptor and lowers its final content in the film. SVD affords precise control over the vertical D/A gradient via coupled solubility–volatility effects during film formation, consistent with our in-situ absorption and GTSAXS analyses of pre-aggregation and depth-dependent structure. Under SVD-0 (toluene only), the high solubility of toluene for L8-BO suppresses

pre-aggregation, keeping L8-BO well dispersed, while its relatively low volatility prolongs drying, enabling L8-BO to infiltrate the D18 nanofiber network and migrate toward the bottom. Consequently, the bottom-region D/A ratio approaches 1:1 (Fig. 3e), increasing the likelihood of acceptor–anode contact and interfacial recombination, which degrades device performance. In contrast, under SVD-120, the benzene fraction is substantially higher. Benzene's lower solubility promotes larger L8-BO pre-aggregates, and its high volatility accelerates drying; together these effects restrict L8-BO penetration into the donor network, yielding limited donor–acceptor intermixing and a consistently low acceptor fraction throughout the depth. Among all conditions, SVD-20 strikes the optimal balance. With a moderate benzene fraction, L8-BO forms appropriately sized pre-aggregates and experiences sufficient—but not excessive—penetration time. As a result, a moderate amount of L8-BO enters the interstices of the D18 network without pronounced bottomward migration, establishing a well-graded vertical D/A profile (Fig. 3f). In line with our GIWAXS/GISAXS/GTSAXS results, this profile supports multiscale domain features that increase D/A interfacial area for exciton dissociation while maintaining pathways conducive to charge transport, and it mitigates electrode-adjacent recombination. Although Mix-0 and Mix-20 yield broadly similar overall distributions, they lack the finely tuned vertical gradient realized by SVD-20, consistent with weaker pre-aggregation signatures and smaller CCLs.

Exciton generation maps (Supplementary Fig. 21) and optical simulations (Supplementary Fig. 22) further corroborate SVD-20's advantage[42,43]. The peak exciton generation occurs near the film center

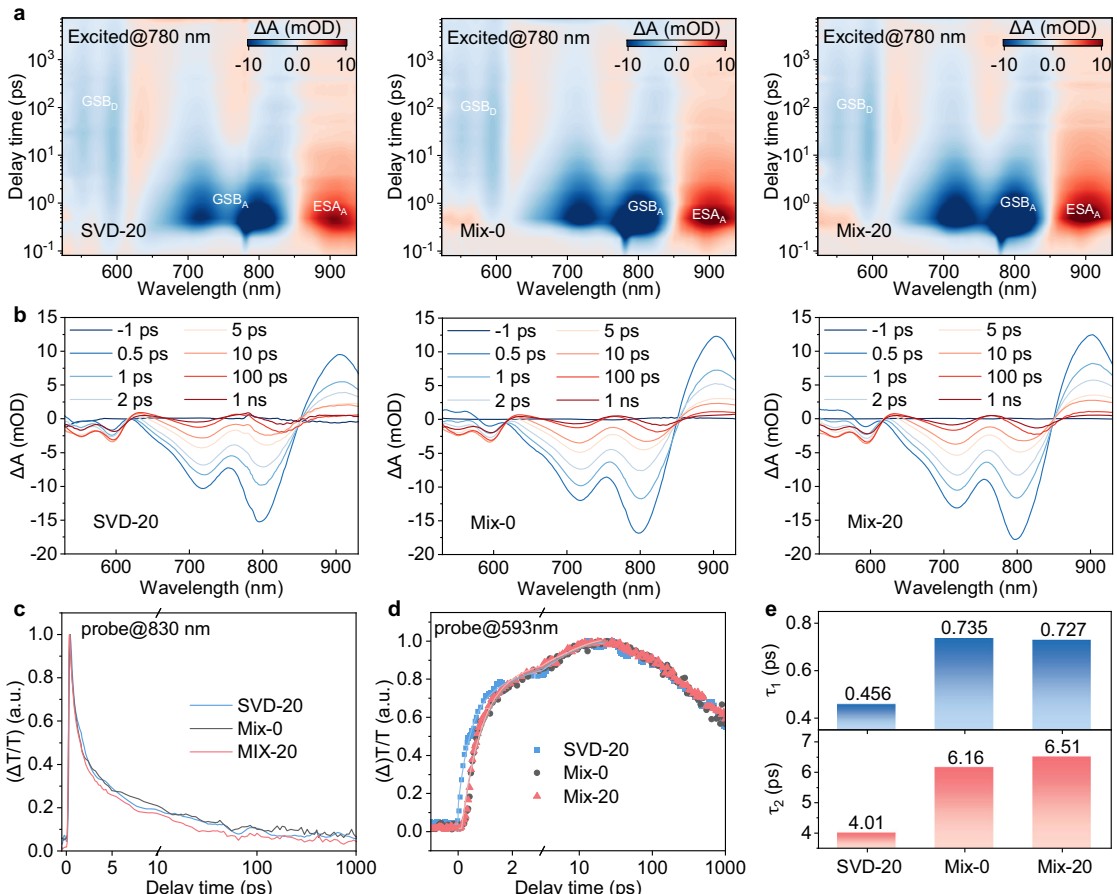

**Fig. 4 | Charge generation dynamics. a** Color plots the fs-TA spectra of the D18/L8-BO films processed under different conditions. **b** Corresponding TA spectra obtained from panels **a**. **c** TA kinetics probed at 830 nm. **d** TA kinetics probed at 593 nm. **e** Hole transfer times estimated from fitting the TA kinetics at 593 nm.

at approximately 57 nm for SVD-20, compared to less favorable depths for SVD-0 (~90 nm, deeper) and SVD-120 (~46 nm, shallower). Mix-0 (~66 nm) and Mix-20 (~65 nm) are intermediate yet still suboptimal. The corresponding film thicknesses are reported in Supplementary Fig. 23. Centralized exciton generation in SVD-20 shortens average transport paths to the respective electrodes, reducing bulk recombination losses. Collectively, these results show that an optimized SVD duration—specifically SVD-20—constructs an advantageous vertical D/A gradient via gradient-driven pre-aggregation and controlled intermixing, thereby enhancing charge transport and suppressing recombination, which directly underpins the device performance of SVD-20.

### Charge generation, transport and recombination

Femtosecond transient absorption spectroscopy (fs-TAS) was used to investigate the impact of SVD on charge generation dynamics (Fig. 4). Upon 780 nm excitation, the acceptor's ground-state bleaching (GSB) and excited-state absorption (ESA) near ~800 nm appeared promptly and then decayed rapidly. Concurrently, a pronounced donor GSB at ~590 nm rose and subsequently decayed, indicating singlet-exciton dissociation and the formation of free charges (polarons) within the active layer. Kinetic traces at 593 nm were fitted with a biexponential model to deconvolute a fast component $\tau_1$ (exciton dissociation at D/A interfaces) and a slower component $\tau_2$ (exciton diffusion). For SVD-20, $\tau_1/\tau_2 = 0.46/4.01$ ps, lower than those of Mix-0 (0.74/6.16 ps) and Mix-20 (0.73/6.51 ps). These results suggest more rapid hole transfer from the acceptor to the donor and more efficient exciton diffusion in SVD-20.

Charge transport properties were quantified via the space-charge-limited current (SCLC) method (Supplementary Fig. 24; Supplementary

Table 9). The hole mobilities ($\mu_e$) for SVD-0, SVD-20, SVD-120, Mix-0, and Mix-20 were $3.5 \times 10^{-4}$, $9.04 \times 10^{-4}$, $2.19 \times 10^{-4}$, $7.22 \times 10^{-4}$, and $7.31 \times 10^{-4}$ cm$^2$ V$^{-1}$ s$^{-1}$, respectively. Correspondingly, their electron mobilities ($\mu_h$) were $5.54 \times 10^{-4}$, $8.13 \times 10^{-4}$, $3.22 \times 10^{-4}$, $3.39 \times 10^{-4}$, and $6.51 \times 10^{-4}$ cm$^2$ V$^{-1}$ s$^{-1}$. These data indicate that the multiscale phase morphology optimized by SVD substantially enhances carrier transport, supporting improved device performance.

Transient photocurrent (TPC) measurements yielded charge-extraction times of 0.28, 0.22, 0.29, 0.25, and 0.32 µs for SVD-0, SVD-20, Mix-0, Mix-20, and SVD-120, respectively (Supplementary Fig. 25a). Transient photovoltage (TPV) revealed charge lifetimes of 6.82, 8.40, 6.27, 7.44, and 5.73 µs for the same sequence (Supplementary Fig. 25b). The conjunction of a longer lifetime and shorter extraction time in SVD-20 signifies more efficient charge extraction with reduced recombination, directly contributing to the photovoltaic performance of D18/L8-BO devices.

The efficiencies of charge generation and collection were further assessed from the dependence of photocurrent density ($J_{ph}$) on the effective voltage ($V_{eff}$) (Supplementary Fig. 26). Two key parameters were extracted: the probability of exciton dissociation ($P_{diss}$) and charge collection efficiency ($P_{coll}$). SVD-20 exhibited the highest $P_{diss}$ of 97.2% (vs. 96.2% for SVD-0, 95.7% for SVD-120, 96.2% for Mix-0 and 96.1% for Mix-20), demonstrating the effectiveness of SVD in promoting exciton dissociation. The $P$coll for SVD-20 reached 89.9% (vs. 88.3% for SVD-0, 87.2% for SVD-120, 89% for Mix-0, and 89.8% for Mix-20), indicating a lower recombination loss in the SVD-20 device.

Light intensity ($P_{light}$) dependent analyses further elucidate recombination pathways. The slope of $V_{OC}$ versus $\ln P_{light}$ gives $nkT/q$, where $n$ is the ideality factor; $n > 1$ indicates trap-assisted (Shockley–Read–Hall)

recombination. SVD-20 shows the lowest $n = 1.05$ (vs. 1.269 for SVD-0, 1.211 for SVD-120, 1.207 for Mix-0, and 1.134 for Mix-20) (Supplementary Fig. 27a), implying suppressed trap-assisted recombination. Additionally, $J_{SC} \propto P_{light}^{\alpha}$ yields higher $\alpha$ of 0.960 for SVD-20 (vs. 0.952 for SVD-0, 0.935 for SVD-120, 955 for Mix-0 and 0.957 for Mix-20) (Supplementary Fig. 27b), indicating efficient charge generation with minimal bimolecular recombination. Taken together, these results demonstrate that SVD-20 enables ultrafast, efficient charge generation, rapid and relatively balanced carrier transport, and suppressed recombination—outcomes consistent with the SVD-induced vertical gradient and multiscale morphology—and ultimately deliver device performance.

### Universality and device stability

To evaluate the universality of SVD, we first replaced the benzene/toluene solvent pair with dioxane/chlorobenzene for processing L8-BO on D18. SVD-30 of L8-BO in dioxane/chlorobenzene increased the PCE from 18.36% to 19.46% (Supplementary Fig. 28; Supplementary Table 10). We also applied the benzene/toluene SVD system to four additional D18-based blends with distinct NFAs (m-TEH, BTP-C6, eC9-2F2Cl, and L8-BO-C4 in Supplementary Fig. 29). For D18/m-THE, SVD-20 increased the PCE from 18.94% to 19.56% (Supplementary Fig. 30; Supplementary Table 11), mirroring the trend observed for L8-BO. Using the same processing conditions, and without blend-specific optimization of solvents or SVD time, the PCE was consistently improved across all systems: from 19.23% to 19.92% for D18/BTP-C6, from 19.08% to 19.69% for D18/eC9-2F2Cl, and from 20.03% to 20.71% for D18 (1% PYIT)/L8-BO-C4 (Supplementary Fig. 31; Supplementary Table 12). Additionally, we replaced D18 with PM6 in the L8-BO system and substituted the benzene/toluene solvent pair with an acetonitrile/toluene pair. After SVD-10 treatment, the device PCE increased from 18.52% to 19.58% (Supplementary Fig. 32; Supplementary Table 13). Collectively, these results highlight the universality of SVD in enhancing device performance, establishing it as a promising strategy for elevating the performance of a wide range of OSCs.

Device stability is a key criterion for assessing the practical viability of OSCs. The stability SVD-0 and SVD-20 processed D18/L8-BO devices were monitored by maximum power point (MPP) tracking at room temperature under continuous white-light illumination (Supplementary Fig. 33). The SVD-20 device exhibits enhanced operational stability, retaining 80% of its initial PCE after approximately 490 h of continuous illumination, whereas the SVD-0 device reaches 80% of its initial PCE after only ~300 h, demonstrating the additional benefit of SVD processing for long-term device performance.

## Discussion

In this work, we developed and implemented an SVD strategy to precisely regulate the multiscale pre-aggregation behavior of NFAs in LbL-OSCs. By establishing a vertical solvent gradient through benzene vapor diffusion into a toluene solution of L8-BO, this approach enables fine-tuned morphological control that significantly enhances device performance, achieving an optimal PCE of 20.18% for D18/L8-BO devices after 20 min SVD. Our comprehensive characterization shows that SVD induces a top-to-bottom gradient in acceptor pre-aggregation that is faithfully translated into multiscale domain structures within the solid film. The resulting hierarchical morphology provides abundant D/A interfaces for efficient exciton dissociation alongside continuous charge-transport pathways, thereby optimizing both exciton separation and carrier collection. In parallel, SVD enables fine regulation of the vertical D/A distribution, positioning the peak exciton-generation zone (~57 nm) near the center of the active layer to shorten carrier transport distances.

Mechanistic investigations confirm that SVD-20 delivers ultrafast charge generation, enhanced and balanced carrier mobilities, and suppressed recombination losses. The enhanced operational stability observed for SVD-20 arises from SVD-induced increases in crystallinity

and more compact, ordered molecular stacking, which stabilize charge-transport pathways and mitigate microstructural degradation during prolonged operation. The optimized vertical acceptor gradient further regulates interfacial density to prevent recombination accumulation and reduces deleterious electrode–active-layer reactions.

Remarkably, this SVD approach exhibits strong universality across diverse material systems and solvent environments. Beyond the model D18/L8-BO system, we successfully applied this strategy to additional NFA-based blends—including BTP-C6, m-TEH, eC9-2F2Cl, and L8-BO-C4—as well as alternative solvent pairs and donor systems, all showing substantial performance gains without system-specific optimization. Notably, the PCE of 20.71% was achieved in the D18 (1% PYIT)/L8-BO-C4 system, among the highest efficiencies reported for single-junction OSCs.

Notwithstanding its strengths, SVD operates within a practical application window defined by solvent and process constraints. It requires fully miscible solvent pairs: the main solvent should offer high solubility for the acceptor and lower volatility than the vapor solvent, while the vapor solvent provides moderately lower solubility to induce controlled pre-aggregation. Key parameters—such as solution height (balancing diffusion length and aggregate size) and tightly regulated temperature—must also be carefully managed. Within this space, we observe robust improvements across several NFAs. Outside it—for example, with very low acceptor solubility, minimal solubility contrast, or poorly controlled thermal conditions—the SVD effect may weaken or produce oversized aggregates that degrade performance. Clarifying these boundaries provides a practical guide to where SVD is most effective and how to tune conditions when extending the method to other D/A and solvent systems.

Building on these clarified operating boundaries, SVD can be applied deliberately to harness gradient-driven pre-aggregation for multiscale microstructure control while avoiding conditions that diminish its benefits. Overall, this work establishes SVD as a robust and versatile method for multiscale microstructure engineering in OSCs, offering a perspective for morphology control through gradient-driven pre-aggregation and advancing the development of high-performance organic electronics.

## Methods

### Materials

D18 and L8-BO were purchased from Solarmer Materials Inc. Ph-4PACz was purchased from VIZUCHEM Co., Ltd. PNDIT-F3N, m-TEH, eC9-2F2Cl were purchased from eflexPV Inc. BTP-C6 was synthesized in our laboratory as reported elsewhere[44]. All the other chemicals were purchased from Aladdin, Adamas, Sigma-Aldrich, and Alfa Asear Chemical Co., and used without further purification.

### Device fabrication

All the devices were fabricated using a structure comprising indium tin oxide (ITO)-coated glass/Ph-4PACz (~1 nm)/donor (~60 nm)/acceptor (~60 nm)/PNDIT-F3N ( ~ 10 nm)/Ag (80 nm). Ph-4PACz was first deposited onto ITO substrates by spin-coating its solution (0.3 mg/mL in ethanol) at 3,000 rpm for 30 s, followed by annealing at 85 °C for 5 min. The donor layer was then prepared by spin-coating a D18 solution (5 mg/mL in chloroform), PM6 solution (10.5 mg/mL in chlorobenzene) or a mixture of D18 (5 mg/mL in chloroform) and 1% PYIT, onto the substrates at 3000 rpm for 30 s. Acceptor materials were dissolved in toluene (15 mg/mL) with 100 wt% DIB as an additive. L8-BO and m-TEH solutions were stirred at 20 °C (± 2 °C) for 30 min, whereas BTP-C6, eC9-2F2Cl, and L8-BO-C4 solutions were stirred at 60 °C (± 2 °C) for 30 min and then allowed to cool to 20 °C (± 2 °C). For each trial, 500 μL of the acceptor solution was transferred into small, lidless glass vials (15 mm in diameter and 17 mm in height). These vials were then placed inside a larger sealed glass bottle (60 mm in diameter and 30 mm in height, sealed with petroleum jelly) containing

3 mL of benzene for the SVD process, ensuring a consistent liquid level and contact area between the solution and the vial wall. The water and oxygen content within the glove box is less than 0.01 ppm. After the required SVD time (0–120 min), the acceptor solutions were gently shaken to ensure uniformity and then deposited onto the pre-formed donor layers to form the active layers, followed by thermal annealing at 85 °C for 5 min. A PNDIT-F3N interlayer (~10 nm) was then deposited by spin-coating its solution (1 mg/mL in methanol with 0.5 v/v% acetic acid) at 2,500 rpm for 30 s. Finally, devices were transferred into the vacuum chamber (~$10^{-5}$ Pa) for Ag electrode deposition (80 nm). The active area of each device was 4 mm$^2$.

### Device characterization

$J$–$V$ measurements were performed using a computer-controlled Keithley 2400 source meter under AM 1.5 G illumination (100 mW cm$^{-2}$) using a solar simulator (XES-70S1, SAN-EI), which was calibrated by a standard Si solar cell (AK-200, Konica Minolta, Inc.). EQE spectra were measured with an EQ-R solar quantum efficiency test system (Enlitech Co., Ltd., Taiwan, China). UV-vis-NIR spectra were obtained on a Shimadzu UV3600i spectrophotometer. AFM images were obtained by using a Bruker Inova atomic microscope in tapping mode.

### Simulation method

The SVD simulation was performed using COMSOL Multiphysics 6.2. The "Transport of Diluted Species" interface within the Chemical Species Transport module was employed to simulate the effect of benzene diffusion into toluene in the small lidless vial used in the SVD setup, specifically tracking the concentration change of benzene within the toluene solution. The simulation was conducted under environmental conditions of 20 °C and $1.013 \times 10^3$ Pa. The simulation was based on the time-dependent (transient) equation:

$$\frac{\partial c_i}{\partial t} + (\nabla \cdot \mathbf{J}_i) + \mathbf{u} \cdot \nabla c_i = R_i \tag{1}$$

where $c_i$ denotes the concentration of the $i$-th species, $t$ represents time, $\mathbf{J}_i$ is the diffusion flux described by Fick's Second Law, $\mathbf{u}$ is the fluid velocity, and $R_i$ is the chemical reaction rate. According to Fick's Law:

$$\mathbf{J}_i = - D_i \cdot \nabla c_i \tag{2}$$

where $D_i$ is the diffusion coefficient, and $\nabla c_i$ is the concentration gradient. For each simulation, a 2D rectangular model (15 mm in length and 3 mm in height) was established to represent the toluene phase. To simplify the boundary conditions, the entire domain was defined as liquid toluene at 20 °C, $1.013 \times 10^3$ Pa. The upper boundary was set as a benzene liquid phase under the same temperature and pressure conditions, with a benzene concentration of 11.2 mol/L, thereby approximating pure benzene. Considering actual vapor diffusion, the diffusion coefficient was set to $3.02 \times 10^{-10}$ m$^2$/s.

### SCLC measurements

The electron-only device with the structure of ITO/ZnO/Active layer/PNDIT-F3N/Ag and the hole-only device with the structure of ITO/Ph-4PACz/Active layer/MoO$_3$/Ag were fabricated. The SCLC measurements employ the Mott-Gurney equation: $J = 9\varepsilon_0\varepsilon_r\mu V^2/8L^3$ to estimate the electron and hole mobilities from dark $J$–$V$ curves obtained from these devices. Here, $J$ is the current density, $\varepsilon_r$ is the relative dielectric constant of the active layer, $\varepsilon_0$ is the permittivity of free space ($8.85 \times 10^{-14}$ F/cm), $\mu$ is the charge mobility, and $L$ is the thickness of the active layer. $V$ is the applied voltage. The $\varepsilon_r$ parameter is assumed to be 3, which is a typical value for organic materials.

### TPV and TPC measurements

TPV and TPC measurements were conducted using the Paios carrier measurement system (Fluxim AG, Switzerland). For TPV, the settling time was 10 ms, the pulse length was 1 ms, and the follow-up time was 10 ms. For TPC, the settling time was 5 μs, the pulse length was 5 μs, and the follow-up time was 50 μs.

### FLAS measurements

FLAS was conducted by Shaanxi Puguangweishi Technology Co., Ltd (Shaanxi, China). Incremental etching of the films was carried out using low-pressure (less than 20 Pa) oxygen plasma. Following each etching step, the UV–vis-NIR absorption spectra were recorded with an optical spectrometer. The thickness-dependent absorption data were analyzed using Beer-Lambert's law to fit the depth-resolved profiles. Detailed methodologies for both measurement and numerical fitting are described elsewhere[45].

### 2D GIWAXS measurements

2D GIWAXS measurements (Pilatus 2 M detector) were carried out at Shanghai Synchrotron Radiation Facility in China (the incidence angle is 0.2°, beam energy is 10 keV, and exposure time is 120 s). The sample to detector distance (SDD) was set to 78 mm. 1D GIWAXS patterns were corrected to represent real $q_r$ and $q_{xy}$ axes with the consideration of missing wedge. The critical incident angle was determined by the maximized scattering intensity from sample scattering with negligible contribution from underneath layer scattering. The samples for the GIWAXS test were prepared by casting the solution onto silicon wafer substrates (ca. 15 × 15 mm2). The data were processed and analyzed by the Nika software package. Gaussian peak fitting was used to obtain peak position and full width at half maximum of the scattering peak. Packing distances can be calculated using $d = 2\pi/q$, where $q$ is the corresponding peak position. CCL was calculated from GIWAXS line cut by the Scherrer equation:

$$CCL = \frac{0.89\lambda}{FWHM \times \cos\left(\arcsin\left(\frac{q\lambda}{4\pi}\right)\right)} \tag{3}$$

Where FWHM is full width at half maximum of the scattering peak, and $\lambda$ is the wavelength of X-rays, which is 0.116 nm.

### GISAXS measurements

GISAXS measurements (Pilatus 2 M detector) were carried out at the Shanghai Synchrotron Radiation Facility in China (the incidence angle is 0.4°). The samples for GISAXS measurements were fabricated on silicon substrates using the same recipe as for the devices. The sample to detector distance (SDD) was set to 2080 mm for GISAXS measurement. 1D GISAXS profiles were extracted on the basis of Debye-Anderson-Brumberger (DAB) and Fractal models, which is presented in Eq. 4:

$$I(q) = \frac{A_1}{\left[1 + (q\xi)^2\right]^2} + A_2(P(q, R))S(q, R, \eta, D) + B \tag{4}$$

The first term of Eq. 4 assigned to DAB is used for simulating the scattering of polymer-rich domain, in which $\xi$ is the average correlation length of the polymer domain, $q$ is the scattering wave vector, and $A_1$ is an independent fitting parameter. The second term of the equation is assigned to the Fractal model, which means the occupation of the fractal-like structure of the non-fullerene acceptor. $P(q, R)$ and $S(q, R)$ are the form factor and fractal structure factor, respectively. The correlation length of the fractal-like structure represents by $\eta$. Guinier radius ($R_g$) is used to characterize the average domain size of acceptor

phase (see Eq. 5).

$$R_g = \sqrt{\frac{D(D+1)}{2}}\eta \qquad (5)$$

## GTSAXS measurements

GTSAXS measurements were performed using a Xeuss 3.0 system. The scattered X-rays were captured by a Dectris EIGER2 Si 1 M photon-counting detector. To prepare a thin-film sample suitable for GTSAXS characterization, the film deposited on a silicon substrate was carefully cleaved from the central region to produce a smooth and clean edge. This step is essential, as the edge quality significantly influences the collection of transmitted scattering signals. The front edge of the sample was positioned at the center of the goniometer stage, aligned with the rotation axis of the incident angle to ensure measurement accuracy[40]. Optical alignment was then performed before X-ray exposure, followed by experimental procedures identical to those used for GISAXS measurements. The incidence angle of 0.08° was first calibrated using the positions of the direct and reflected beams on the detector, conducted with a short exposure time and a SDD of 2 m. Subsequently, GTSAXS measurements were carried out at incidence angles of 0.12°, 0.16°, and 0.20°, respectively.

## fs-TAS measurements

The fs-TAS measurements were performed using a Helios setup (Ultrafast Systems) equipped with a Ti:sapphire regenerative amplifier laser system (Coherent Libra) delivering 780 nm laser pulses (100 fs, 1 kHz). The probe beam was produced by focusing a portion of the fundamental femtosecond laser beam onto either a sapphire plate or a yttrium aluminum garnet (YAG) plate to cover the visible and near-infrared spectral ranges, respectively. All TAS results are presented in units of ΔOD, where negative features correspond to GSB or stimulated emission (SE), and positive features indicate ESA. During the TA measurements, the samples were kept under a nitrogen atmosphere to prevent photodegradation. The pump fluence was maintained at 2 µJ/cm$^2$ to minimize exciton–exciton annihilation.

## Reporting summary

Further information on research design is available in the Nature Portfolio Reporting Summary linked to this article.

# Data availability

The data that support the findings of this study are available within the article and its Supplementary Information/Source Data file. Source data are provided with this paper.

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

## Acknowledgements

This work was financially supported by the National Key Research and Development Program of China (Grant No. 2022YFB4200500), National Natural Science Foundation of China (NSFC, 22379101, 22375133, and 22422904), Sichuan Natural Science Foundation (2024NSFSC0001). The authors thank Yanping Huang from Center of Engineering Experimental Teaching, School of Chemical Engineering, Sichuan University for the help of gas chromatography measurements.

## Author contributions

X.X. and Q.P. conceived the project. X.X. and W.Z. designed the experiments. W.Z. fabricated and characterized the cells. X.D., B.F. and H.L. helped perform the GIWAXS, GISAXS. X.X. drafted the original manuscript. W.Z., Y.W., and Q.P. helped with the manuscript preparation. X.X. and Q.P. supervised the project. All the authors discussed the results and commented on the manuscript.

## Competing interests

The authors declare no competing interests.
