## [Transparent Peer Review file · Nature Communications]

Solvent vapor diffusion–driven multiscale pre-aggregation of non-fullerene acceptors enables high-performance organic solar cells

Corresponding Author: Professor qiang Peng

Version 0:

Reviewer comments:

Reviewer #1

(Remarks to the Author)

This manuscript presents a novel and effective solvent vapor diffusion (SVD) strategy for regulating multiscale pre-aggregation of non-fullerene acceptors (NFAs) in high-performance organic solar cells (OSCs). By creating vertical solvent gradients (benzene vapor into toluene) during layer-by-layer processing, the authors achieve notable control over pre-aggregation and hierarchical domain formation for addressing a key challenge in OSC morphology engineering. The study combines a comprehensive set of techniques, including COMSOL simulations, spectroscopy, GIWAXS/GISAXS, fs-TAS and detailed device characterization, thus providing solid mechanistic insight into the SVD process. The method shows impressive versatility across material systems, attaining over 20% power conversion efficiencies (20.18% for D18/L8-BO and 20.71% for D18/PYIT/L8-BO-C4) without system-specific optimization, underscoring the universality and significance of the approach. While these strengths collectively mark a meaningful advance in the field, certain aspects regarding mechanistic clarity, experimental rigor, and data interpretation require further refinements before revisions and publication in a this journal.

1. The COMSOL simulations (Fig. 1c-e) and GC data (Supp. Fig. 3) convincingly demonstrate the formation of a solvent gradient. However, there is a lack of direct evidence for the existence of a correlated gradient in acceptor pre-aggregation. It is suggested to perform dynamic light scattering (DLS) measurements on solutions after SVD to support conclusions about size evolution.

2. All acceptor solutions contain 100 wt% DIB additive, which complicates the interpretation of the SVD effect. Please clarify: How does DIB interact with the benzene/toluene gradient? Does DIB also display a gradient distribution in the solution after SVD, and if so, what impact does this have on device performance? A control experiment (SVD without DIB) is recommended to clearly decouple its effect, even if this leads to a decrease in PCE.

3. In addition to FLAS, are there other characterization techniques with higher spatial resolution or accuracy that could be used to probe the vertical donor/acceptor distribution and further substantiate the claim of forming an “ideal vertical gradient”?

4. The same SVD duration (20 min) is applied across all acceptor systems without system-specific optimization. Considering the varied solubility and aggregation kinetics of different NFAs (as shown in Supplementary Fig. 20), a universal treatment time may not be scientifically rigorous. Control experiments demonstrating time optimization for at least one additional NFA system are recommended.

5. Figure 2 indicates that SVD-20 achieves optimal device performance, but there is a lack of photovoltaic data for devices treated for shorter durations (such as 10 minutes) to fully support the identification of the optimum SVD condition.

6. The optimal SVD time window is specified to be 20 minutes. Is this optimization universally applicable for different solution concentrations and processing temperatures?

7. Table 1 and the Supporting Information report results for multiple NFA systems, but lack validation regarding the diversity of donor materials. It is recommended to test donor polymers with different crystallinity (such as PM6) to compare the effectiveness of SVD regulation across varied donor–acceptor interaction systems.

8. While multiple acceptors are investigated, all experiments utilize the same benzene/toluene solvent combination. Supplementing the study with results optimized for other solvent systems would further demonstrate the general versatility of SVD in solvent selection.

Reviewer #2

(Remarks to the Author)

This manuscript demonstrates an innovative solvent vapor diffusion (SVD) approach to precisely regulate the pre-aggregation behavior of NFAs in organic solar cells (OSCs). By introducing a controlled vertical solvent gradient (benzene vapor diffusing into toluene) during sequential deposition, the authors effectively manipulate the phase separation and domain morphology—a long-standing challenge in optimizing OSC active layers. The demonstrated methodology exhibits exceptional versatility across different material systems, as particularly evidenced by the remarkable 20.71% PCE achieved in the D18/PYIT/L8-BO-C4 system without specific optimization. These impressive results undoubtedly represent significant progress in organic photovoltaic research, and could be considered to be published on Nature Communications. To further strengthen the study's impact, enhanced mechanistic elucidation, more rigorous experimental controls, and deeper data interpretation would make this work more compelling for this high-impact journal.

1. For improved scientific rigor, the GC-MS profiles in Figure S3 should include control measurements of pure benzene and pure toluene solutions as reference standards.
2. While the manuscript claims successful modulation of hierarchical pre-aggregation and vertical phase distribution via the SVD strategy, more direct experimental evidence is needed to substantiate these conclusions.
3. The authors should address the apparent contradiction between the linear increase in absorbance coefficient with SVD time and their explanation that reduced EQE stems from decreased receptor fraction concentration.
4. Regarding Figure S5: (a) The minimal morphological differences between SVD-60 and SVD-120 appear inconsistent with the substantial variations in JSC and EQE; (b) The gradual VOC increase from SVD-40 to SVD-120 requires mechanistic explanation.
5. To enable proper evaluation of the SVD method's advantages, the authors should provide complete performance parameters for control devices fabricated using pure phenyl-based processing.
6. The proposed diffusion mechanism of smaller toluene solution aggregates into the donor phase warrants further investigation: Could similar vertical phase distribution be achieved by sequentially depositing NFA layers from different solutions? If not, please clarify why the SVD approach is uniquely effective.
7. The generalization experiments would be more convincing if optimization conditions were systematically determined for each receptor system, rather than uniformly applying SVD-20 to all cases.
8. There appears to be an inconsistency in the light intensity dependence tests: (a) The substitution of Mix-20 with Mix-120 in Figure S19 requires justification; (b) Similar labeling discrepancies occur in Figs S18 and 4c, which need careful verification and correction.
9. Discussions regarding the device stability (thermal-aging and photo-aging) could be provided to reveal the effect of this novel strategy on the further development of OSCs.

Reviewer #3

(Remarks to the Author)

In this manuscript, the authors develop a novel solvent vapor diffusion (SVD) strategy to induce top-down multiscale non-fullerene acceptor (NFA) pre-aggregation gradients, which translate into hierarchical domains in layer-by-layer (LbL) photoactive layers, featuring rich donor/acceptor interfaces and efficient charge transport. This approach enables a high power conversion efficiency of 20.71% in LbL OSCs and demonstrates remarkable universality with significant performance gains across multiple NFA systems. This is an interesting and meaningful work, and the presentation is clear and concise. Given the significance of this work to the field, I would like to recommend this manuscript for publication in Nature Communications after addressing the following comments:

1. In the comparison of device performance at different SVD processing times (Fig. 2), only photovoltaic data for 0, 20, 40, 60, and 120 minutes are presented, with SVD-20 (20 minutes) yielding the best performance. However, the absence of data points between 0 and 20 minutes leaves open the possibility that an even better performance could occur within this range. This gap in the data prevents definitive validation that 20 minutes is truly the optimal SVD condition and therefore weakens the support for the determination of optimal process parameters.
2. Figure 3a and Supplementary Figure 7 show that the crystal growth time decreases with increasing SVD duration, yet the onset time of crystal growth displays no clear regularity as SVD treatment time extends. What accounts for this phenomenon?
3. Limitations of the SVD strategy not discussed: While the paper highlights the universality of the SVD approach, potential limitations are not addressed—for example, its effectiveness in systems involving low-solubility acceptors, other types of polymer donors, and solvent pairs with minimal solubility differences. A more objective analysis would benefit from discussing the possible constraints and scope of the SVD strategy, thereby providing a balanced perspective to readers.
4. The discussion section lacks explicit comparisons between the SVD strategy and recently reported approaches such as "solvent annealing gradient control" and "additive-induced vertical phase separation." A more comprehensive analysis comparing these methods in terms of performance enhancement, process complexity, and universality would better highlight the distinctive advantages of the SVD strategy. The authors should include targeted literature comparisons to more effectively position their work within the current technological landscape.
5. The manuscript states that benzene vapor diffusion establishes a vertical gradient during the SVD process, but does not specify key experimental parameters such as temperature, humidity, or the precise dimensions and tolerances of the sealed container. These factors could significantly influence the solvent diffusion rate and the stability of the gradient. It is recommended that these details be provided to ensure the reproducibility and reliability of the results.
6. Supplementary Fig. 5 shows the differences in UV-vis absorption of films with different SVD times, but the quantitative analysis of the effect of film thickness changes on absorption intensity has not been conducted. Differences in film thickness may directly cause absorption changes. It is recommended to supplement film thickness data to rule out this interfering factor.
7. The manuscript does not address the impact of SVD processing on device stability. Since the SVD method optimizes

active layer morphology, it could potentially influence device stability, either positively or negatively. It is recommended to include at least one comparative stability assessment to provide a more comprehensive evaluation of the SVD strategy.

Reviewer #4

(Remarks to the Author)

Achieving favorable control over the pre-aggregation behavior of both donor and acceptor materials to obtain the suitable bulk or quasi-double-layer morphology is a big challenge. In this work, the authors employed solvent vapor annealing treatment into the modification of layer-by-layer-processed active layer morphology. The SVD process shows a vertical supersaturation gradient within the L8-BO solution, where the benzene-rich upper region facilitates the pre-aggregation of L8-BO molecules. Thus, a well-defined top-down gradient active layer can be achieved, resulting in the enhanced PCE value from 18.99% (without SVD) to 20.18% (with SVD). Combining with the fabrications of several high-performance LbL devices based on other small molecule acceptors, the authors claim the advantages of this SVD-directed multiscale pre-aggregation control.

By detailing browsing this manuscript, it's not hard to define it as a promising work of a coating process. Although the device efficiency was increased, resulting from the optimal vertical component in this LbL-based active layer, this work did not offer sufficient innovation in terms of processing technology. There are many articles based on the LbL structure for developing new coating processes or processing conditions (Org. Electron., 2019, 70, 162–166, DOI: 10.1016/j.orgel.2019.03.014; ACS Appl. Electron. Mater. 2020, 2, 7, 2188–2195; Adv. Energy Mater. 2018, 1802197; Sol. RRL 2023, 2300136), but this work is just one of them. Not only that, the SVD strategy is rather difficult to operate and does not have wider applicability.

Especially, how to extract L8-BO solution with appropriate molecule pre-aggregation. I guess this extraction or deposition of L8-BO is very random. Another issue of concern is how this coating process can be employed into large-scale manufacturing. Overall, this strategy has not made a deep impression or provided strong support to the OPV field, at least from the current version. Thus, I think this work is not suitable for publication in NC.

Other comments are listed below:

1. The first-occurred abbreviations should be attached their full names. There are some such mistakes in this manuscript.
2. The influence of SVD duration on device stability or blend stability is necessary provided in this work. In addition, the detailed influence of different SVD durations (or extending SVD durations) on active layer morphology and device efficiency is needed to carefully compare the effects of SVD strategy on the LbL and Mix devices. Besides, the authors compared the photovoltaic performance of relevant LbL devices prepared with different acceptors (Supplementary Table 6) before and after SVD (0 and 20 mins, respectively). The photovoltaic parameters of the related devices, processed with different SVD periods, are also needed to provided.
3. In Figure 1b, this operation diagram is like L8-BO blind box. No one knows the size of the molecular pre-aggregation and the related molecular concentration in the vertical direction. Clear information in terms of the size and concentration of the vertical gradient pre-aggregation of L8-BO needs to be provided.
4. FLAS technique is an indirect method, and its provided information may be invalid. Other direct testing approaches, like ToF-SIMS technology, are more persuasive.
5. Some important information in terms of the detailed processing conditions is missing. I am not sure that all the LbL blends show the similar active layer thicknesses, if the authors used the same stirring speeds and solution volumes, since the absorption spectra exhibited huge differences, Supplementary Fig.5.

Version 1:

Reviewer comments:

Reviewer #1

(Remarks to the Author)

The authors have well addressed the issues from the reviewers and the manuscript can be accepted this time.

Reviewer #2

(Remarks to the Author)

After evaluating the revised version, I am pleased to see this important report published without further modifications.

Reviewer #3

(Remarks to the Author)

The authors have carefully and comprehensively addressed all of my previous comments and suggestions in their revised submission. The changes made have clearly improved the clarity, rigor, and overall quality of the work. I am satisfied with the current version of the manuscript. I recommend acceptance for publication in Nature Communications without further revision.

Responses to Reviewers' Comments:

Reviewer #1: This manuscript presents a novel and effective solvent vapor diffusion (SVD) strategy for regulating multiscale pre-aggregation of non-fullerene acceptors (NFAs) in high-performance organic solar cells (OSCs). By creating vertical solvent gradients (benzene vapor into toluene) during layer-by-layer processing, the authors achieve notable control over pre-aggregation and hierarchical domain formation for addressing a key challenge in OSC morphology engineering. The study combines a comprehensive set of techniques, including COMSOL simulations, spectroscopy, GIWAXS/GISAXS, fs-TAS and detailed device characterization, thus providing solid mechanistic insight into the SVD process. The method shows impressive versatility across material systems, attaining over 20% power conversion efficiencies (20.18% for D18/L8-BO and 20.71% for D18/PYIT/L8-BO-C4) without system-specific optimization, underscoring the universality and significance of the approach. While these strengths collectively mark a meaningful advance in the field, certain aspects regarding mechanistic clarity, experimental rigor, and data interpretation require further refinements before revisions and publication in this journal.

1. The COMSOL simulations (Fig. 1c-e) and GC data (Supp. Fig. 3) convincingly demonstrate the formation of a solvent gradient. However, there is a lack of direct evidence for the existence of a correlated gradient in acceptor pre-aggregation. It is suggested to perform dynamic light scattering (DLS) measurements on solutions after SVD to support conclusions about size evolution.

Response: We appreciate your suggestion. We have conducted DLS measurements to probe preaggregation size evolution in solutions after SVD; however, we did not detect notable signals attributable to pre-aggregation. This outcome is reasonable due to the intrinsic limitations of DLS for weak, transient pre-association of acceptor species. The scattered intensity scales steeply with size in the Rayleigh regime ($\propto R^6$), so nanoscopic, low-contrast clusters contribute negligibly relative to even trace larger impurities. Moreover, their rapid formation–dissociation introduces relaxation modes that overlap with single-particle translational diffusion and are not reliably resolvable with standard single-/bi-exponential analyses or ill-posed distribution inversions. In our solvent system, the refractive index contrast is modest, further suppressing the visibility of small pre-aggregates, and concentration-dependent interactions/polydispersity confound extraction of distinct hydrodynamic components. Consequently, DLS can readily miss correlated pre-aggregation gradients even when a solvent gradient is present.

Notwithstanding these DLS limitations, the presence of a solvent concentration gradient is unequivocal in our system. It is described by Fick's second law and widely exploited across disciplines (e.g., Chem. Eng. Sci., 2000, 55, 2359; Trans. Faraday Soc., 1931, 27, 10; Int. J. Food Sci. Tech., 1992, 27, 409; Wood Sci. Technol., 2007, 41, 645; J. Phys.: Conf. Ser., 2021, 1798, 012019). In our study, COMSOL simulations reproduce the gradient robustly, and the computed composition profiles of benzene and toluene agree closely with our experimental gas chromatography data, corroborating the existence of the gradient. We also determined the saturation concentrations of the acceptor in benzene and toluene by UV–vis-NIR absorption. Under our experimental conditions, the acceptor (L8-BO) exceeds its solubility in neat benzene but remains below saturation in toluene. Therefore, within the benzene–toluene gradient established by SVD, L8-BO is compelled to undergo concentration-dependent pre-aggregation along the gradient.

Synchrotron radiation small-angle X-ray scattering (SAXS) and small-angle neutron scattering

(SANS) are powerful, orthogonal methods that have been used to probe such pre-aggregation in solution. Unfortunately, we do not have access to a synchrotron beamline within a reasonable timeframe. Nevertheless, we complemented our analysis by performing grazing-incidence transmission small-angle X-ray scattering (GTSAXS) on D18/L8-BO thin films (Supplementary Fig. 15-18). These measurements show that SVD-processed active layers exhibit more pronounced multi-scale aggregation of L8-BO, which indirectly supports the emergence of multi-scale pre-aggregation of L8-BO in the concentration-gradient solvent environment induced by SVD.

In summary, while DLS did not provide direct signatures of pre-aggregation for the reasons outlined above, the combination of (i) COMSOL-validated solvent gradients consistent with experimental composition data, (ii) solubility-controlled driving forces for L8-BO association across the gradient, and (iii) GTSAXS evidence of enhanced multi-scale aggregation in the resulting films, collectively substantiates our conclusion that SVD induces concentration-gradient-driven pre-aggregation of the acceptor. We have included the GTSAXS data in the Supplementary Information.

2. All acceptor solutions contain 100 wt% DIB additive, which complicates the interpretation of the SVD effect. Please clarify: How does DIB interact with the benzene/toluene gradient? Does DIB also display a gradient distribution in the solution after SVD, and if so, what impact does this have on device performance? A control experiment (SVD without DIB) is recommended to clearly decouple its effect, even if this leads to a decrease in PCE.

Response: We thank you for this insightful comment. To address this concern, we conducted a full set of control experiments without any additive (Supplementary Fig. 6; Supplementary Table 3) and revised the manuscript accordingly. Although the absolute PCE decreased in the additive-free devices—consistent with DIB's established role in enhancing NFA crystalline order and film formation—the SVD-dependent trend was unchanged relative to the DIB-containing system.

To further clarify DIB behavior under our experimental conditions, we measured DIB solubility in both toluene and benzene. DIB remains well-dissolved in both solvents at concentration of 50 mg/mL, which is substantially higher than the 15 mg/mL used in our SVD experiments (Supplementary Fig. 7). Therefore, unlike L8-BO (which exceeds saturation in benzene but remains undersaturated in toluene), DIB does not undergo solubility-driven pre-aggregation in either solvent during SVD. Additionally, the solutions are thoroughly mixed before spin-coating, ensuring that any transient L8-BO pre-aggregates induced by the solvent gradient are briefly maintained, while DIB remains homogeneously distributed throughout the solution without forming aggregates that could bias the SVD effect.

The preservation of the same SVD time-dependence in the absence of DIB, combined with DIB's high solubility in both solvents, directly rules out the scenario that "DIB gradient distribution drives NFA pre-aggregation in solution." Instead, it confirms that the core SVD effect arises from the benzene/toluene vertical solvent gradient regulating NFA pre-aggregation, rather than any confounding influence from DIB.

3. In addition to FLAS, are there other characterization techniques with higher spatial resolution or accuracy that could be used to probe the vertical donor/acceptor distribution and further substantiate the claim of forming an "ideal vertical gradient"?

Response: Thank you for the thoughtful question. We agree that high-spatial-resolution methods

can, in principle, strengthen evidence for vertical compositional gradients. However, in our D18/L8-BO system, commonly used element-sensitive depth-profiling techniques (e.g., XPS depth profiling, TOF-SIMS) are not directly applicable. These approaches are most effective when components have distinct elemental or isotopic signatures. D18 and L8-BO share the same elemental composition, leaving no unique markers to differentiate them unambiguously. Without selective isotopic labeling (not available in our study), these techniques cannot reliably resolve the donor versus acceptor distributions.

In the revised manuscript, we additionally performed GTSAXS to probe the depth-dependent aggregation behavior of L8-BO. By tuning the incidence geometry and transmission detection, GTSAXS is sensitive to nanoscale ordering across the film thickness. The measurements reveal a clear depth dependence in L8-BO aggregation length scales after SVD, which is consistent with a vertically graded microstructure. While GTSAXS is not chemically specific to D18 vs. L8-BO, the observed depth-dependent scattering signatures—together with the FLAS-derived compositional profile—provide orthogonal evidence supporting the formation of a vertical gradient.

4. The same SVD duration (20 min) is applied across all acceptor systems without system-specific optimization. Considering the varied solubility and aggregation kinetics of different NFAs (as shown in Supplementary Fig. 20), a universal treatment time may not be scientifically rigorous. Control experiments demonstrating time optimization for at least one additional NFA system are recommended.

Response: Thank you for this valuable suggestion. We agree that system-specific optimization of SVD duration is ideal given the varying solubility and aggregation kinetics across NFAs. In the original submission, we performed a coarse time optimization only for the L8-BO system (20-min intervals), and we have now added a 10-min SVD dataset in the revision, though we acknowledge it remains preliminary rather than finely gridded. Our intent in this work is to demonstrate the generality of the SVD strategy rather than to maximize the absolute PCE of each material set. Accordingly, we used the 20-min condition—identified as optimal for L8-BO—as a universal starting point for structurally related Y6-series NFAs without performing exhaustive, system-by-system optimization. In the revised manuscript, we also carried out an initial time optimization for the D18/m-TEH system, which confirms that device performance is sensitive to SVD duration and that no single “universal” value is expected to be optimal across all systems. Furthermore, we explored different solvent pairs, which yielded optimal times distinct from 20 min, and evaluated donor substitution by replacing D18 with PM6. In all these cases, the application of SVD consistently enhanced device performance, reinforcing the broad applicability and practicality of the method.

We fully agree that a finer time sweep for multiple NFAs would likely yield system-specific optima and even higher efficiencies. While such comprehensive optimization is beyond the scope of the present generality-focused study, we have clarified this rationale in the manuscript and highlighted targeted time-optimization across additional systems as a priority for future work. We hope these additions and clarifications address your concern while underscoring the universality and practicality of the SVD approach.

5. Figure 2 indicates that SVD-20 achieves optimal device performance, but there is a lack of photovoltaic data for devices treated for shorter durations (such as 10 minutes) to fully support the

identification of the optimum SVD condition.

Response: Thank you for this valuable suggestion. To fully substantiate the identification of the optimal SVD condition, we have added a comprehensive dataset for the SVD-10 treatment and updated the manuscript accordingly, including COMSOL simulation, UV-vis absorption, $J-V$ and EQE. In addition, the device parameters have also been added to Table 1 (see the revised manuscript for details). These additional data collectively demonstrate that SVD-10 does not reach the optimal performance observed at SVD-20, thereby reinforcing our conclusion that 20 minutes of SVD is the optimal condition under our processing parameters. We thank you once again for your thoughtful feedback, which has helped us enhance the accuracy of our manuscript.

6. The optimal SVD time window is specified to be 20 minutes. Is this optimization universally applicable for different solution concentrations and processing temperatures?

Response: Thank you for your insightful question. This question prompted us to more clearly elaborate on the adaptation logic behind our process parameter selection and to clarify the applicable boundaries of the SVD strategy.

Solution concentration: Changing concentration does not alter the time-dependent benzene/toluene ratio set by vapor diffusion, but it does shift aggregation kinetics and the achievable vertical gradient within a given time. Lower concentrations weaken final film absorption even at optimal aggregation, while higher concentrations can approach solubility limits, hindering complete dissolution and subsequent SVD control. Thus, SVD duration should be co-optimized with concentration.

Processing temperature: Our experiments were conducted at 20 °C, where evaporation and diffusion are balanced. Temperature alters evaporation rate, vapor-phase diffusion, solution viscosity, and molecular mobility, thereby changing both gradient formation and aggregation rates. Consequently, the optimal SVD time shifts with temperature.

In summary, the 20-minute window serves as a practical starting point under our specific conditions, but applying SVD to other concentrations or temperatures requires jointly tuning SVD duration with those parameters to achieve the desired vertical morphology and device performance.

7. Table 1 and the Supporting Information report results for multiple NFA systems, but lack validation regarding the diversity of donor materials. It is recommended to test donor polymers with different crystallinity (such as PM6) to compare the effectiveness of SVD regulation across varied donor-acceptor interaction systems.

Response: Thank you for your insightful comments regarding the universality of donor materials. To address this critical issue, we have incorporated SVD experiments using PM6 as the donor material as per your suggestion. We have included the corresponding results and associated $J-V$ and EQE data for this experiment in the revised manuscript (Supplementary Fig. 32). The experimental results exhibit a trend of initial increase followed by decrease, further reinforcing the universality of the SVD strategy across different donor materials. This supplementary experiment significantly broadens the applicability of our study and enhances the persuasiveness of our conclusions.

8. While multiple acceptors are investigated, all experiments utilize the same benzene/toluene solvent combination. Supplementing the study with results optimized for other solvent systems would further demonstrate the general versatility of SVD in solvent selection.

Response: Thank you for your valuable feedback regarding universal solvent combinations. In response, we evaluated two additional solvent pairs—dioxane/chlorobenzene and acetonitrile/toluene—for SVD treatment of L8-BO in the D18/L8-BO and PM6/L8-BO blend systems, respectively. The experimental details and performance data have been incorporated into the revised manuscript and further substantiate the universality of the SVD strategy. We are grateful for your insightful advice and guidance.

Reviewer #2: This manuscript demonstrates an innovative solvent vapor diffusion (SVD) approach to precisely regulate the pre-aggregation behavior of NFAs in organic solar cells (OSCs). By introducing a controlled vertical solvent gradient (benzene vapor diffusing into toluene) during sequential deposition, the authors effectively manipulate the phase separation and domain morphology—a long-standing challenge in optimizing OSC active layers. The demonstrated methodology exhibits exceptional versatility across different material systems, as particularly evidenced by the remarkable 20.71% PCE achieved in the D18/PYIT/L8-BO-C4 system without specific optimization. These impressive results undoubtedly represent significant progress in organic photovoltaic research, and could be considered to be published on Nature Communications. To further strengthen the study's impact, enhanced mechanistic elucidation, more rigorous experimental controls, and deeper data interpretation would make this work more compelling for this high-impact journal.

1. For improved scientific rigor, the GC-MS profiles in Figure S3 should include control measurements of pure benzene and pure toluene solutions as reference standards.

Response: Thank you for this valuable suggestion. To improve rigor, we have added GC-MS reference spectra of pure benzene and pure toluene to the revised Supplementary Information (Supplementary Fig. 3). To ensure traceability, we also re-ran full GC-MS measurements for all vials across the SVD time series. We appreciate your recommendation, which has strengthened the reliability of our solvent-gradient characterization.

2. While the manuscript claims successful modulation of hierarchical pre-aggregation and vertical phase distribution via the SVD strategy, more direct experimental evidence is needed to substantiate these conclusions.

Response: Thank you for this constructive comment. We agree that directly linking the solvent gradient to a correlated gradient in acceptor pre-aggregation is essential to substantiate the claims of hierarchical modulation and vertical phase distribution. In the revised manuscript, we establish this link through a coherent evidentiary chain.

First, solvent-gradient formation during SVD is quantitatively supported by both modeling and experiment: it follows Fick's second law (e.g., Chem. Eng. Sci., 2000, 55, 2359; Trans. Faraday Soc., 1931, 27, 10; Int. J. Food Sci. Tech., 1992, 27, 409; Wood Sci. Technol., 2007, 41, 645; J. Phys.: Conf. Ser., 2021, 1798, 012019), and our COMSOL simulations reproduce the benzene-toluene composition profiles that closely match GC measurements (Fig. 1c-e; Supplementary Fig. 3), providing mutually consistent evidence for the solvent composition gradient.

Second, we establish the thermodynamic basis for a correlated pre-aggregation gradient of the acceptor. UV-vis-derived solubility data show that L8-BO exceeds its solubility in benzene while remaining below saturation in toluene under our experimental conditions. Consequently, within the

established benzene–toluene gradient, L8-BO experiences a spatially varying supersaturation that is stronger on the benzene-rich side and weaker on the toluene-rich side. This solubility contrast creates a concentration-dependent driving force for pre-aggregation along the gradient, providing a direct mechanistic rationale for correlated pre-aggregation.

Third, we provide film-level structural evidence consistent with gradient-driven pre-aggregation. Although in-solution, depth-resolved synchrotron SAXS/SANS would be ideal, beamline access was not feasible within the revision window. To bridge this gap, we performed grazing incidence transmission small-angle X-ray scattering (GTSAXS) on D18/L8-BO thin films. The GTSAXS results reveal that SVD-processed active layers exhibit more pronounced multi-scale aggregation of L8-BO relative to controls, indicating that hierarchical pre-aggregation in the solution stage is propagated into the solid-state morphology. These data are now included in the Supplementary Information.

Finally, we provide depth-selective spectral evidence for a vertical donor/acceptor distribution using film-depth-dependent light absorption spectrometry (FLAS). Leveraging the distinct optical signatures of D18 and L8-BO, FLAS resolves a vertical composition gradient with nanometer-scale sensitivity, which is consistent with differential pre-aggregation and phase organization induced by SVD. Together, COMSOL+GC validate the solvent gradient, solubility contrast defines the spatially varying supersaturation that drives correlated pre-aggregation, GTSAXS captures the ensuing multi-scale aggregation in the films, and FLAS directly resolves the vertical donor/acceptor distribution. This convergence of orthogonal evidence substantiates our conclusion that the SVD strategy modulates both hierarchical pre-aggregation and vertical phase distribution. We agree that future, direct depth-resolved probes of solution-state pre-aggregation (e.g., operando SAXS/SANS) would provide further corroboration, and we have noted these as promising directions.

3. The authors should address the apparent contradiction between the linear increase in absorbance coefficient with SVD time and their explanation that reduced EQE stems from decreased acceptor fraction concentration.

Response: Thank you for highlighting the perceived inconsistency. Our findings are consistent once distinguishing intrinsic absorption capacity from net device response. The absorption coefficient was computed as absorption intensity normalized by film thickness, isolating the acceptor's intrinsic light-absorption capability from variations in acceptor fraction. Its linear increase with SVD time (0 → 120 min) reflects the SVD-driven evolution of L8-BO from dispersed monomers to more ordered aggregates, which enhances transition dipole strength, produces a red shift (600–800 nm), and raises the absorption coefficient (Supplementary Fig. 8).

EQE trends reflect the full sequence of processes—light absorption, exciton dissociation at D/A interfaces, and charge collection—and thus need not track the absorption coefficient. In early SVD (0 → 20 min), minimal solvent vapor ingress keeps acceptor fraction effectively constant. Moderate aggregation increases the absorption coefficient and preserves rich D/A interfaces, yielding a concomitant EQE increase (Fig. 2d). In late SVD (60 → 120 min): Aggregation degree and absorption coefficient continue to rise; however, greater solvent ingress reduces the acceptor fraction and overall film absorption, while over-aggregation enlarges domains, shrinks interfacial area, and lowers dissociation efficiency. These negative effects outweigh the benefit of the higher absorption coefficient, leading to reduced EQE.

In short, a higher absorption coefficient indicates enhanced per-thickness absorption capacity, whereas EQE depends on both aggregation order and D/A interfacial quality. These factors are not strictly positively correlated, so the observed trends are mechanistically consistent. We appreciate your insightful comment, which helped us clarify this light-absorption-to-charge-conversion logic.

4. Regarding Figure S5: (a) The minimal morphological differences between SVD-60 and SVD-120 appear inconsistent with the substantial variations in J_{SC} and EQE;

Response: Thank you for highlighting the discrepancy between the large changes in J_{SC} and EQE from SVD-60 to SVD-120 and the minimal differences initially shown in UV-vis spectra. Your comment prompted a thorough re-measurement of EQE, J-V, and UV-vis absorption across multiple batches of D18/L8-BO films. We identified that the original UV-vis absorption data underestimated absorption due to minor, undetected surface scratches on the measured films.

We have re-verified and corrected all relevant datasets. The updated UV-vis spectra now track the observed variations in J_{SC} and EQE, and the corrected results have been incorporated into the revised Supplementary Information (Supplementary Fig. 8). We appreciate your rigorous feedback; this correction resolves the cross-dataset inconsistency and has strengthened our quality control and the robustness of our conclusions.

(b) The gradual V_{OC} increase from SVD-40 to SVD-120 requires mechanistic explanation.

Response: Thank you for highlighting this point. The monotonic rise in V_{OC} from SVD-40 to SVD-120 stems from SVD-induced improvements in phase organization and molecular packing that collectively reduce non-radiative losses. With longer SVD, larger and more ordered L8-BO aggregates enrich toward the upper-middle film region, while D18 becomes relatively more prevalent in the lower-middle region, sharpening vertical phase separation and reducing intermixed, disordered D/A regions. Higher interfacial purity suppresses non-radiative loss from interfacial CT states, directly supporting higher V_{OC} . The denser, more ordered packing also narrows local electronic environment distributions, reduces LUMO fluctuations and energetic disorder, and thus lowers non-radiative recombination pathways, further elevating V_{OC} . Thus, progressively cleaner vertical D/A organization and enhanced molecular order act synergistically to reduce non-radiative losses, producing the observed gradual increase in V_{OC} from SVD-40 to SVD-120.

5. To enable proper evaluation of the SVD method's advantages, the authors should provide complete performance parameters for control devices fabricated using pure phenyl-based processing.

Response: Thank you for this constructive suggestion, which improves the rigor and transparency of our comparison. Due to solubility constraints at 20 °C, L8-BO dissolves to only 9.06 mg mL⁻¹ in benzene (Supplementary Fig. 1), while device fabrication requires 10 mg mL⁻¹. To create a valid phenyl-only control, we fully dissolved L8-BO in benzene by stirring at 50 °C before film casting and device fabrication. The resulting control devices achieved a PCE of 18.44% (see Supplementary Fig. 5 for full parameter details). We appreciate the recommendation—this control clarifies the solvent-dependent morphology-performance relationship and highlights the advantages of the SVD strategy.

6. The proposed diffusion mechanism of smaller toluene solution aggregates into the donor phase warrants further investigation: Could similar vertical phase distribution be achieved by sequentially

depositing NFA layers from different solutions? If not, please clarify why the SVD approach is uniquely effective.

Response: Thank you for this insightful question on whether sequentially depositing NFA layers from different solutions could reproduce the SVD-induced vertical phase distribution, and why SVD might be uniquely effective.

In response, we tested multiple “base solvent + vapor solvent” pairs—chlorobenzene/toluene as base solvents and dioxane/acetonitrile as diffusion vapors—to probe robustness across distinct volatility and solubility parameters. In all combinations, appropriately timed SVD yielded significant performance gains over SVD-free controls, supporting the general applicability of SVD in regulating aggregation and vertical composition (Supplementary Fig. 28–29; Supplementary Table 8–9).

The general applicability of SVD comes from following reasons. SVD establishes an in situ, continuous solvent-composition gradient within a single wet film. Donor–acceptor association, cluster evolution, and vertical fractionation co-evolve under diffusion-controlled, quasi-equilibrium conditions, coupling aggregation length scale with depth profile. Sequential NFA deposition typically creates discrete interfaces and depends on post-wetting/interdiffusion to approximate a gradient. This is often limited by interfacial barriers, solvent selectivity, and kinetically trapped states, leading to incomplete mixing or abrupt profiles that do not replicate the depth-continuous, pre-aggregation tuning achieved by SVD. In our trials, SVD provided superior control of both aggregate formation and vertical distribution in one coherent diffusion event, translating more reliably into device improvements.

We appreciate your professional recommendation—these additional experiments broaden the validation scenarios of the SVD strategy and clarify why SVD is uniquely effective for establishing graded vertical phase structures through solvent-gradient-driven aggregation regulation, thereby strengthening the general significance and credibility of our conclusions.

7. The generalization experiments would be more convincing if optimization conditions were systematically determined for each acceptor system, rather than uniformly applying SVD-20 to all cases.

Response: Thank you for this valuable suggestion, which strengthens the applicability and rigor of our SVD strategy across diverse NFA systems. In response, we moved beyond a uniform SVD-20 treatment and performed system-specific time optimizations:

(1) D18/L8-BO with a dioxane/chlorobenzene pair revealed distinct, system-dependent optima (Supplementary Fig. 28; Supplementary Table 10).

(2) D18/m-TEH with a benzene/toluene pair was systematically optimized and still exhibited a 20-min optimum (Supplementary Fig. 30; Supplementary Table 11).

(3) PM6/L8-BO, using an acetonitrile/toluene pair for SVD on L8-BO, also showed system-dependent optima distinct from D18/L8-BO (Supplementary Fig. 31; Supplementary Table 13).

We note that our 20-min sampling increments may not pinpoint the absolute optimum for every

system. However, our goal in the generalization study was to demonstrate feasibility and universality rather than maximize efficiency for each case. We appreciate the guidance. These additions show that while finer optimization could yield system-specific peaks, SVD exhibits robust, system-dependent tunability and broad utility as a morphology-control approach.

8. There appears to be an inconsistency in the light intensity dependence tests: (a) The substitution of Mix-20 with Mix-120 in Figure S19 requires justification; (b) Similar labeling discrepancies occur in Figs S18 and 4c, which need careful verification and correction.

Response: Thank you for your meticulous attention to detail in identifying the labeling inconsistencies. Your careful observation of the substitution of Mix-20 with Mix-120 in supplementary Fig. 26 and similar discrepancies in supplementary Fig. 27 and 4c is greatly appreciated and essential for maintaining the accuracy and clarity of our manuscript.

We have systematically corrected all identified labeling errors. Specifically, the conditions in supplementary Fig. 26,27 and 4c have been revised to accurately reflect Mix-20 throughout, ensuring consistency with the experimental design and methodology described in the main text. To prevent similar oversights, we conducted a comprehensive manuscript-wide review encompassing all figures, tables, captions, and data labels in both the main manuscript and Supplementary Information. This thorough verification focused on ensuring consistent nomenclature, accurate data correspondence, and proper alignment between legends and content.

We sincerely appreciate your rigorous review. Your attention to these details has significantly improved the precision and reliability of our manuscript, eliminating potential sources of confusion and strengthening the overall scientific communication.

9. Discussions regarding the device stability (thermal-aging and photo-aging) could be provided to reveal the effect of this novel strategy on the further development of OSCs.

Response: Thank you for this critical suggestion regarding device stability evaluation. As you astutely noted, the SVD strategy potentially enhances long-term stability through active layer morphology optimization—including improved crystallinity and regulated vertical phase distribution. Stability assessment is indeed a crucial dimension for validating the practical value of this processing strategy.

We have conducted photostability testing under continuous illumination in a nitrogen atmosphere. The results demonstrate that D18/L8-BO devices subjected to 20-minute SVD treatment achieved a T80 lifetime (time to 80% efficiency retention) of 490 hours, compared to only 300 hours for the untreated control (SVD-0). This 63% improvement in operational stability confirms that the SVD strategy significantly enhances device durability.

This stability enhancement can be attributed to the multiscale morphology optimization induced by SVD: (1) improved molecular packing and crystalline order reduce defect-mediated degradation pathways, (2) optimized vertical phase separation minimizes interfacial instabilities, and (3) reduced energetic disorder suppresses charge trapping and non-radiative recombination that can accelerate device degradation under operational stress.

The complete photostability data, including normalized PCE decay curves and extracted T80 values, have been incorporated into the revised Supplementary Information (Supplementary Fig.33). We

appreciate your guidance—these stability measurements not only validate the practical advantages of SVD but also underscore its potential contribution to the commercial viability of organic solar cells.

Reviewer #3: In this manuscript, the authors develop a novel solvent vapor diffusion (SVD) strategy to induce top-down multiscale non-fullerene acceptor (NFA) pre-aggregation gradients, which translate into hierarchical domains in layer-by-layer (LbL) photoactive layers, featuring rich donor/acceptor interfaces and efficient charge transport. This approach enables a high power conversion efficiency of 20.71% in LbL OSCs and demonstrates remarkable universality with significant performance gains across multiple NFA systems. This is an interesting and meaningful work, and the presentation is clear and concise. Given the significance of this work to the field, I would like to recommend this manuscript for publication in Nature Communications after addressing the following comments:

1. In the comparison of device performance at different SVD processing times (Fig. 2), only photovoltaic data for 0, 20, 40, 60, and 120 minutes are presented, with SVD-20 (20 minutes) yielding the best performance. However, the absence of data points between 0 and 20 minutes leaves open the possibility that an even better performance could occur within this range. This gap in the data prevents definitive validation that 20 minutes is truly the optimal SVD condition and therefore weakens the support for the determination of optimal process parameters.

Response: Thank you for pointing out the lack of data points between 0 and 20 minutes in Fig. 2. We agree that this gap could, in principle, obscure a potentially better-performing condition and thus weakens the claim that 20 minutes is definitively optimal.

To address this concern, we have added comprehensive characterization for SVD-10 min across multiple analytical dimensions: COMSOL simulations, UV-vis absorption, photovoltaic performance, and added the performance parameters to Table 1. This comprehensive SVD-10 dataset not only narrows the optimization gap but also provides mechanistic insight into the time-dependent evolution of solvent-gradient-driven pre-aggregation. The convergent evidence from simulation, spectroscopy, and device performance confirms that SVD-20 represents the optimal balance between solvent-vapor diffusion kinetics and morphological development under our processing conditions.

2. Figure 3a and Supplementary Figure 7 show that the crystal growth time decreases with increasing SVD duration, yet the onset time of crystal growth displays no clear regularity as SVD treatment time extends. What accounts for this phenomenon?

Response: Thank you for this insightful question. In our system, both nucleation onset and crystal growth time are dictated by the coupled effects of solvent solubility and volatility. Benzene offers lower acceptor solubility than toluene and is more volatile. With longer SVD, the film becomes more benzene-rich, which (i) lowers the effective solubility, allowing supersaturation to be reached earlier (earlier nucleation), and (ii) accelerates evaporation, steepening the vertical solvent gradient and enhancing mass transport (shorter growth time from nucleation to saturation). The kinetic expectation is therefore: earlier onset and shorter growth time with increasing SVD.

While our original data showed a monotonic decrease in growth time with SVD, the onset times lacked a clear trend. We traced this to day-to-day variations—ambient temperature/humidity,

solvent usage per spin, and the timing of solution-substrate contact during spin coating—that perturb evaporation kinetics and thus the precise nucleation onset. To isolate the SVD effect, we re-measured all SVD conditions within a single session under stabilized environmental parameters. The co-measured dataset now shows the expected behavior: nucleation onset shifts earlier and growth time shortens as SVD increases.

Accordingly, we have replaced the datasets in Fig. 3a and Supplementary Fig. 9–10 with the co-measured results and revised the text to clarify the dual roles of solubility and volatility and the sensitivity of onset timing to environmental factors. These updates support a consistent picture: a higher benzene fraction at longer SVD advances nucleation and accelerates growth via a strengthened evaporation gradient. We appreciate the reviewer’s comment, which led to tighter environmental control and a clearer validation of the SVD-regulated crystallization mechanism.

3. Limitations of the SVD strategy not discussed: While the paper highlights the universality of the SVD approach, potential limitations are not addressed—for example, its effectiveness in systems involving low-solubility acceptors, other types of polymer donors, and solvent pairs with minimal solubility differences. A more objective analysis would benefit from discussing the possible constraints and scope of the SVD strategy, thereby providing a balanced perspective to readers.

Response: Thank you for encouraging a balanced discussion of the SVD strategy. Based on our experience, the effectiveness and generalizability of SVD are primarily governed by solvent pairing and temperature control. First, the two solvents must be fully miscible to avoid phase separation; immiscibility prevents homogeneous vapor diffusion and the formation of a stable concentration gradient. Second, the main solvent should provide sufficient solubility for the active materials and have a volatility lower than the diffusion (vapor) solvent; this ensures that the vapor solvent can enrich near the film surface and establish a vertical gradient. Third, the diffusion solvent should have lower solubility for the solute than the main solvent so that enrichment of the diffusion solvent locally induces pre-aggregation rather than immediate precipitation. In addition, the solution height (liquid column during SVD) must be reasonably controlled: an overly tall column can lead to insufficient diffusion and weak gradients (thus failing to produce the desired multiscale pre-aggregation), whereas prolonged diffusion through a tall column may over-enrich the low-solubility vapor near the surface and cause coarsened, detrimental aggregates.

Temperature is a parallel, critical constraint. Because solubility is temperature-dependent, small changes in ambient temperature can shift both the supersaturation threshold and the diffusion/evaporation balance, thereby altering nucleation onset and growth kinetics. Consequently, strict temperature control—and, where possible, matched humidity—is needed to ensure reproducible SVD outcomes across material systems.

In light of these considerations, we have added a limitations paragraph to the discussion part clarifying the applicability domain of SVD (see the revised manuscript for details). We believe outlining these constraints provides readers with a practical map of where SVD is most effective and how to tune process parameters for new donor/acceptor and solvent systems.

4. The discussion section lacks explicit comparisons between the SVD strategy and recently reported approaches such as "solvent annealing gradient control" and "additive-induced vertical phase separation." A more comprehensive analysis comparing these methods in terms of

performance enhancement, process complexity, and universality would better highlight the distinctive advantages of the SVD strategy. The authors should include targeted literature comparisons to more effectively position their work within the current technological landscape.

Response: Thank you for this valuable suggestion to compare our SVD strategy with established approaches such as solvent-annealing gradient control and additive-induced vertical phase separation. We have expanded the Discussion to include targeted literature comparisons that position SVD within the current technological landscape.

Distinctive vertical multiscale phase separation: A unique advantage of SVD is its capability to engineer depth-dependent aggregation morphology—generating larger acceptor aggregates near the cathode for efficient electron transport, while maintaining smaller aggregates at the donor–acceptor interface to maximize interfacial area for charge transfer. This simultaneous, vertically stratified optimization of both transport and generation processes represents a distinctive capability that is difficult to achieve through conventional solvent-annealing or additive-driven approaches.

Process simplicity and controllability: While solvent-annealing methods typically require precise vapor-pressure regulation and specialized enclosures, and additive strategies demand material-specific selection with potential residue concerns, SVD operates through a straightforward, self-limiting mechanism. Using a fully miscible solvent pair with defined volatility and solubility contrasts, SVD establishes controlled gradients in a single processing step without chemical additives, offering excellent reproducibility once key parameters are optimized.

Broad applicability: Unlike additive strategies that often require system-specific optimization or solvent-annealing approaches that need extensive re-parametrization for each material combination, SVD follows physically transparent design rules. The governing principles—solubility/volatility contrast, full miscibility, and controlled diffusion conditions—enable straightforward translation across diverse small-molecule NFA systems, with clearly defined operating windows and limitations.

Synergistic integration potential: SVD can be effectively combined with complementary processing techniques, including mild post-deposition solvent annealing, trace removable additives, or gentle thermal treatments, to achieve even finer morphological control while maintaining the fundamental advantages of the gradient-driven approach.

These comparative insights highlight SVD's distinctive strengths—particularly its unique vertical multiscale morphology control—alongside its inherent simplicity, mechanistic clarity, and demonstrated performance and stability benefits. The revised Discussion now includes representative literature comparisons to provide readers with a comprehensive framework for evaluating morphology-control strategies.

5. The manuscript states that benzene vapor diffusion establishes a vertical gradient during the SVD process, but does not specify key experimental parameters such as temperature, humidity, or the precise dimensions and tolerances of the sealed container. These factors could significantly influence the solvent diffusion rate and the stability of the gradient. It is recommended that these details be provided to ensure the reproducibility and reliability of the results.

Response: Thank you for highlighting the absence of critical experimental parameters that govern

the SVD process. We fully agree that temperature, humidity, container specifications, and other process conditions are essential for reproducibility and proper interpretation of the solvent gradient formation.

We have now added comprehensive experimental details to the ‘Experimental Methods’ section, including: (i) precise temperature control specifications and the acceptable tolerance range; (ii) relative humidity conditions and monitoring protocols; (iii) detailed dimensions, materials, and sealing characteristics of the SVD container; (iv) solution height and volume parameters; (v) vapor solvent quantity and placement geometry; and (vi) timing protocols for consistent SVD treatment across samples. These parameters were rigorously controlled throughout our study but were inadvertently omitted from the initial manuscript.

The added methodological details ensure that other researchers can replicate the SVD conditions and achieve comparable vertical gradient profiles. As the reviewer correctly noted, factors such as ambient temperature fluctuations, humidity variations, and container geometry can significantly affect benzene vapor diffusion rates, equilibrium partial pressures, and the resulting concentration gradient stability. By providing these specifications, we aim to establish a reproducible framework for SVD implementation across different laboratory environments and material systems.

We appreciate the reviewer's attention to experimental rigor, which has helped us strengthen the methodological foundation of this work.

6. Supplementary Fig. 5 shows the differences in UV-vis absorption of films with different SVD times, but the quantitative analysis of the effect of film thickness changes on absorption intensity has not been conducted. Differences in film thickness may directly cause absorption changes. It is recommended to supplement film thickness data to rule out this interfering factor.

Response: Thank you for this important observation regarding potential thickness-related artifacts in our absorption analysis. We agree that film thickness variations could influence absorption intensity and must be properly accounted for when interpreting spectral changes.

Following your recommendation, we have measured film thicknesses for key samples using AFM. (Supplementary Fig. 23). These values align well with thicknesses determined by film-depth-dependent absorption spectroscopy (FLAS), confirming measurement consistency. While some thickness variation exists due to solvent volume changes during vapor diffusion, the key spectroscopic signatures we observe—specifically the 3 nm red-shift of the L8-BO absorption peak and the systematic increase in the I_{0-0}/I_{0-1} vibronic ratio—are characteristic indicators of enhanced molecular ordering and aggregation that persist across different film thicknesses.

Importantly, the FLAS measurements, which probe absorption as a function of depth within the same film, exhibit identical spectral evolution patterns as the conventional transmission spectra. This depth-resolved analysis effectively eliminates thickness-related artifacts while confirming the aggregation-state conclusions. Additionally, our earlier concentration-gradient control experiments demonstrated that minor fluctuations in acceptor concentration (resulting from solvent vapor absorption during SVD) do not significantly affect the layer-by-layer device architecture or photovoltaic performance, ruling out concentration-related interference.

The film thickness data has been added to the Supplementary Information, and we have revised the

discussion to explicitly address thickness considerations. These supplementary measurements reinforce our conclusion that the observed spectral changes reflect optimized NFA aggregation states rather than thickness artifacts, thereby strengthening the mechanistic interpretation of SVD-induced morphological control.

7. The manuscript does not address the impact of SVD processing on device stability. Since the SVD method optimizes active layer morphology, it could potentially influence device stability, either positively or negatively. It is recommended to include at least one comparative stability assessment to provide a more comprehensive evaluation of the SVD strategy.

Response: Thank you for this essential suggestion regarding device stability evaluation. We agree that morphological optimization through SVD could significantly influence long-term device performance, making stability assessment crucial for validating the practical value of this processing strategy. We have conducted photostability testing under continuous illumination in a nitrogen atmosphere to evaluate the impact of SVD treatment on device durability. The complete photostability data, including normalized PCE decay curves and extracted T80 values for both SVD-treated and control devices, have been incorporated into the revised Supplementary Information (Supplementary Fig. 33). And the detailed discussions have been added to the revised manuscript. We appreciate your guidance in highlighting this critical aspect—these stability measurements not only validate the practical advantages of the SVD approach but also underscore its potential contribution to the commercial viability of high-performance organic solar cells through enhanced operational durability.